# Simple phosphinate ligands access zinc clusters identified in the synthesis of zinc oxide nanoparticles

Sebastian D. Pike[1], Edward R. White[1], Milo S.P. Shaffer[1] & Charlotte K. Williams[1,†]

The bottom-up synthesis of ligand-stabilized functional nanoparticles from molecular precursors is widely applied but is difficult to study mechanistically. Here we use $^{31}$P NMR spectroscopy to follow the trajectory of phosphinate ligands during the synthesis of a range of ligated zinc oxo clusters, containing 4, 6 and 11 zinc atoms. Using an organometallic route, the clusters interconvert rapidly and self-assemble in solution based on thermodynamic equilibria rather than nucleation kinetics. These clusters are also identified *in situ* during the synthesis of phosphinate-capped zinc oxide nanoparticles. Unexpectedly, the ligand is sequestered to a stable $Zn_{11}$ cluster during the majority of the synthesis and only becomes coordinated to the nanoparticle surface, in the final step. In addition to a versatile and accessible route to (optionally doped) zinc clusters, the findings provide an understanding of the role of well-defined molecular precursors during the synthesis of small (2–4 nm) nanoparticles.

[1] Department of Chemistry, Imperial College London, Imperial College Road, South Kensington Campus, London SW7 2AZ, UK. † Present address: Oxford Chemistry, Chemical Research Laboratory, 12 Mansfield Road, Oxford OX1 3 TA, UK. Correspondence and requests for materials should be addressed to M.S.P.S. (email: m.shaffer@imperial.ac.uk) or to C.K.W. (email: charlotte.williams@chem.ox.ac.uk).

Zinc oxide nanoparticles are used in a huge range of contexts, ranging from wide band gap semiconductors and photo-electronic materials[1–3], to catalysts and photoactive antimicrobial surfaces[4–8]. Individualized nanoparticles are synthesized in solution via a similar broad range of techniques including sol-gel chemistry and hydrolysis/thermolysis routes[9–15]. Regardless of the synthesis method, ZnO and indeed other nanoparticles are frequently capped by surfactants or organic ligands, commonly alkyl amines or carboxylic acids, to maximize solubility during growth and use. In these cases, as with many nanoparticle systems, a ligand is usually implicitly considered to control the synthesis, even if exchanged later[16,17]. Despite the importance of such systems, the detailed mechanism by which discrete, often monometallic, molecular precursors and ligands are combined to form functionalized nanoparticles is not well understood[18,19]. One very attractive approach to ZnO nanoparticle synthesis, pioneered by Chaudret and colleagues[10], hydrolyses organo-zinc reagents, at room temperature in organic solvents, to form crystalline Wurtzite products[12,20]. This route provides small particles and is compatible with thermally sensitive organic/polymer chemistry[21]; it has enabled the preparation of high-performance composite photovoltaic (PV) cells[22,23], colloidal catalysts[20,24] and antimicrobial plastics[4]. In the context of the mechanistic studies reported below, the route provides the opportunity for detailed control of precursor/ligand stoichiometry, as excess ligand can be avoided[12], and the extent of reaction is limited by water supply.

Metallic cluster complexes can grow to more than 1 nm in size and bridge between the domains of molecules and nanoparticulates[25]; they frequently show unusual and impressive optical, electronic and magnetic properties[26,27]. In contrast, the field of zinc-oxo cluster chemistry is relatively less developed, although it is known that alkyl zinc alkoxide/carboxylate clusters are useful precursors to form ZnO nanoparticles and thin films[9,11–13,15,18,28]. Doped materials may also be prepared from heterobimetallic clusters, with improved properties attributed to intimate mixing of the two metals[13,28]. Furthermore, tetrahedral zinc carboxylate clusters, $[Zn_4O(CO_2R)_6]$, are ubiquitous vertices in metal organic frameworks (MOFs), showing outstanding gas sorption and separation characteristics, although they can be sensitive to attack from water or donor solvents[29–33]. The metal-oxygen framework structures of reported precursors do not generally map directly onto the Wurzite–ZnO structure[10,12]; therefore, their transformations into nanoparticulate ZnO probably involves significant molecular rearrangement. In the case of $[RZnOR]_4$ complexes, alkoxide is lost before ZnO nucleation (to a Wurzite structure) and the relationship of the ligands to the growing nanoparticle is not clear. Here we show that ligands may coordinate to cluster species, which act as spectators, while ZnO nucleation occurs and act as a 'ligand reservoir' that is only consumed at the end of the synthesis procedure.

Although the alkyl zinc cluster chemistry using ligands such as alkoxide or carboxylate shows significant promise[13,34–39], utilization of alternative ligands is less well explored. Furthermore, attempts to analyse the speciation during carboxylate/alkoxide precursor transformations result in product mixtures and broadened, complex nuclear magnetic resonance (NMR) spectra. In contrast, ligands coordinated with a P-containing group provide a $^{31}P$ NMR spectroscopic handle that allows for simple identification of individual cluster geometries even when a complicated mixture is present. Various discrete complexes of zinc phosphonate $[RPO_3]^{2-}$ or dialkylphosphate $[(RO)_2PO_2]^-$ are known as viable precursors to nanomaterials[40–42]. One stand-out example is a well-defined $Zn_{12}$ cluster, which contains a $Zn_4O$ core surrounded by $Zn-Et$ fragments supported by eight phosphonate ligands[41]. This cluster is proposed to be a viable precursor to porous zincophosphonate materials. Phosphinate ($[R_2PO_2]^-$) ligands are iso-electronic with carboxylates and could be attractive alternatives. The parent phosphinic acids are more acidic than carboxylic acids (for example, $pK_a$: diphenylphosphinic acid (DPPA-H), 2.3; benzoic acid, 4.2) and thus, although the bonding to zinc is slightly weaker, they should be less susceptible to hydrolysis. Furthermore, having two R groups increases their steric protection and enhances hydrophobicity. Stability to hydrogenation and good solubilizing properties make dioctylphosphinate an interesting ligand for supporting nanoparticles used for quasi-homogeneous hydrogenation catalysis (for example, the hydrogenation of $CO_2$ to MeOH)[20]. Despite their promise as ligands, well-defined zinc complexes and clusters coordinated by monoanionc phosphinate ligands are hardly studied[43].

Here, the reactions of simple, commercially-available diethyl zinc with various equivalents of phosphinic acids and water are used to reproducibly prepare a series of new clusters, which are all fully characterized, including by X-ray diffraction (XRD; Fig. 1). Alongside more typical alkyl zinc ligand and tetrahedral $Zn_4O(ligand)_6$ clusters, some very unusual larger structures are identified, including partially hydrolysed zinc clusters and those containing hydroxy or boroxine cores. Interestingly, these species all show equilibrium relationships with each other and other small molecules, suggesting the clusters may readily interconvert in solution. With the detailed understanding of these species, it is possible to directly identify them *in situ* during the synthesis of ZnO nanoparticles.

## Results

**Synthesis of zinc phosphinate cluster complexes.** The first part of the study focused on understanding and characterizing the species present during simple reactions between diethyl zinc and DPPA-H (as a model ligand). Thus, the reaction between equimolar quantities of $ZnEt_2$ and DPPA-H forms a new tetra-zinc cluster, **1A**. Its $^{31}P\{^1H\}$ NMR spectrum shows a sharp singlet (23.2 p.p.m.) and the $^1H$ NMR spectrum shows a 1:1 ratio of ethyl:DPPA resonances (Supplementary Figs 1 and 2). Although the structures of alkyl zinc phosphinate complexes are not yet reported, alkyl zinc carboxylates adopt a range of chemical structures[44], including hexa-[35] or pentanuclear complexes[37–39]. Crystals of **1A**, analysed by XRD, show a distorted cubic structure $[Zn_4Et_4(DPPA)_4]$ with a tetrahedral arrangement of zinc atoms (Fig. 2 and Supplementary Figs 3,4). Each zinc is singly coordinated to a $P=O$ oxygen ($P=O$ range, 1.492(2)–1.497(2) Å) and each $P-O^-$ oxygen atom ($P–O$ range, 1.531(2)–1.534(2) Å) bridges between two zinc centres. The shape of **1A** is most closely related to the 'cubane' structures of alkyl zinc alkoxides but with the phosphinate ligand adopting bidentate chelation[13].

Compound **1A** is highly moisture sensitive and the addition of $\sim 2$ eq. of water (allowing full hydrolysis of all $Zn-Et$ bonds) forms a new species, **2A**, which also exhibits a single peak in the $^{31}P\{^1H\}$ NMR spectrum (32.9 p.p.m.; Supplementary Figs 5–8). Complex **2A** is also a tetrazinc cluster of the form $[Zn_4(\mu_4-O)(DPPA)_6]$, featuring a central $\mu_4$-oxo atom and six phosphinate ligands (Supplementary Fig. 9); the excess zinc/ethyl is likely to be hydrolysed to form NMR-silent insoluble ZnO particles. Compound **2A** can also be directly prepared, in quantitative yield (NMR spectroscopy), by reaction of a 4:6:1 ratio of $ZnEt_2$:DPPA-H:water, in toluene or $CH_2Cl_2$ (Fig. 3). The matrix-assisted laser desorption/ionization–time of flight (MALDI–ToF) mass spectrum shows a peak for $[Zn_4O(DPPA)_5]^+$ in keeping with the expected cluster formula (Supplementary Fig. 7). The $^1H$ NMR spectrum shows a single environment for the

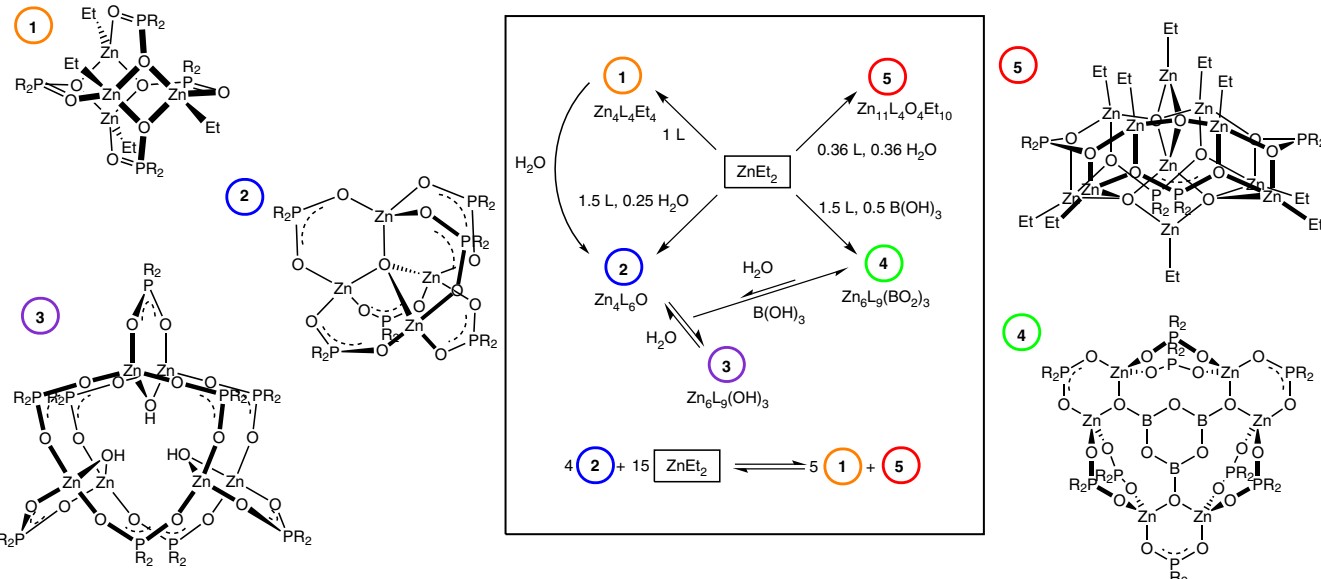

**Figure 1 | Zinc cluster complexes.** The zinc cluster structures and their interconversions, where $L = R_2PO_2H$, $R = Ph$, $C_6H_4OMe$ or $C_8H_{17}$ (**4** only formed with $R = Ph$).

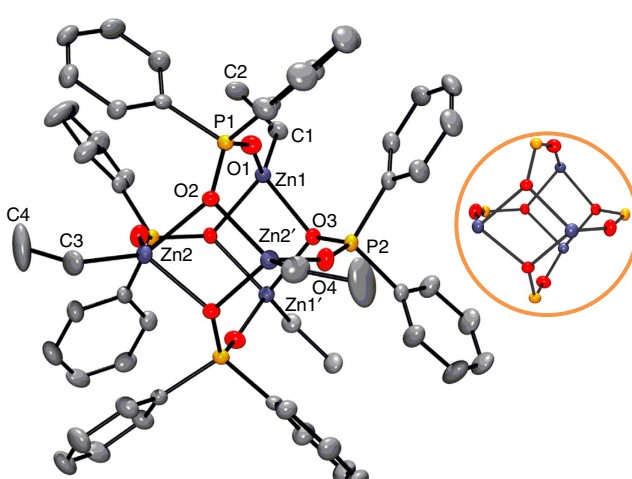

**Figure 2 | Solid-state structure of 1A.** H-atoms omitted for clarity; a view of the Zn cluster core structure, with the phenyl/ethyl groups omitted, is provided within the coloured circle.

phenyl substituents in **2A** (Supplementary Fig. 6). The crystal structures of **2A**, grown either from toluene or $CH_2Cl_2$/hexane, show four independent molecules in the asymmetric unit. The phenyl groups could not be located to acceptable degrees of accuracy; however, $[Zn_4(\mu_4\text{-}O)(O_2P)_6]$ units were observed (Supplementary Figs 9, 10). By using bis(4-methoxyphenyl) phosphinic acid, a fully resolvable crystal structure of **2B** was obtained, a complex with an analogous structure to **2A** (Fig. 3a). The structure of **2B** shows a tetrahedral core of $Zn_4O$ capped by six bidentate ligands, which show equal $P-O$ bond lengths (within error), indicating a delocalized coordination mode (Supplementary Figs 11–15). Although such structures are not yet known for phosphinate ligands, they are commonly observed for other anionic ligands and the benzene dicarboxylate $Zn_4O$ cluster is a common construct in MOFs[29,30,45].

Samples of **2A** exposed to moisture led to the formation of a new complex, **3A**, which displays two $^{31}P$ NMR signals in a 2:1 ratio (30.1 (1P), 24.2 (2P) p.p.m.; Supplementary Figs 16–20).

The addition of 5 eq. (versus **2A**) of water to the solution results in a mixture comprising a relative molar ratio **2A**:**3A** = 2:3. Solvated water is also observed in the NMR spectrum, suggesting equilibration between **2A**, water and **3A** (Fig. 3). It is important to emphasize that **3A** reproducibly forms on addition of water to chloroform, toluene or tetrahydrofuran (THF) solutions of **2A**.

The formula of **3A** is $[Zn_6(\mu_2\text{-}OH)_3(DPPA)_9]$, established by XRD analysis of single crystals (Fig. 3b and Supplementary Fig. 18). Solid **3A** can also be isolated in quantitative yield by direct reaction, in this case of a 1:1.5:0.75 ratio of $ZnEt_2$: DPPA-H:water. The product has a pseudo trigonal prismatic shape, previously unknown for such zinc clusters. It also features three bridging zinc hydroxide ligands. The trigonal prismatic shape results from two planar triangular units of $Zn_3(DPPA)_3$, which are bridged by three further DPPA units and three hydroxides. In all cases, P–O bond lengths are similar, suggesting delocalized bonding.

The stability of the cluster may stem from the hydroxide groups being positioned just far enough from each other to hinder any further condensation reactions (Supplementary Fig. 21). Attenuated total reflection–infrared spectroscopy of a crystalline sample of **3A** (dried under vacuum) shows a weak signal at $3,644\,cm^{-1}$ attributed to O–H stretches (not present for **2A**), consistent with values reported for other bridging $Zn_2(\mu_2\text{-}OH)$ units; a second broad signal at $3,410\,cm^{-1}$ could be a different OH stretching mode or traces of adsorbed moisture (Supplementary Fig. 18)[46–49]. The $^1H$ NMR spectrum of **3A** shows three sets of phenyl environments in a 1:1:1 ratio, which are assigned using correlation spectroscopy NMR (Supplementary Fig. 17). The solution structure is consistent with the solid-state structure, assuming that there is some flexibility enabling a $D_{3H}$ symmetry in solution. The six 'planar' DPPA units are assigned to two phenyl environments: one pointing approximately in the plane of the triangle and the other perpendicular. The third phenyl environment is attributed to the three 'bridging' DPPA units in which both phenyl groups occupy identical environments. The hydroxide protons are also observable as a sharp signal at 3.67 p.p.m., with a relative integral of 3H.

The formation of **3A** is unexpected; the isolation and characterization of well-defined Zn-hydroxide complexes is usually challenging, often requiring the use of bulky, multi-dentate ligands

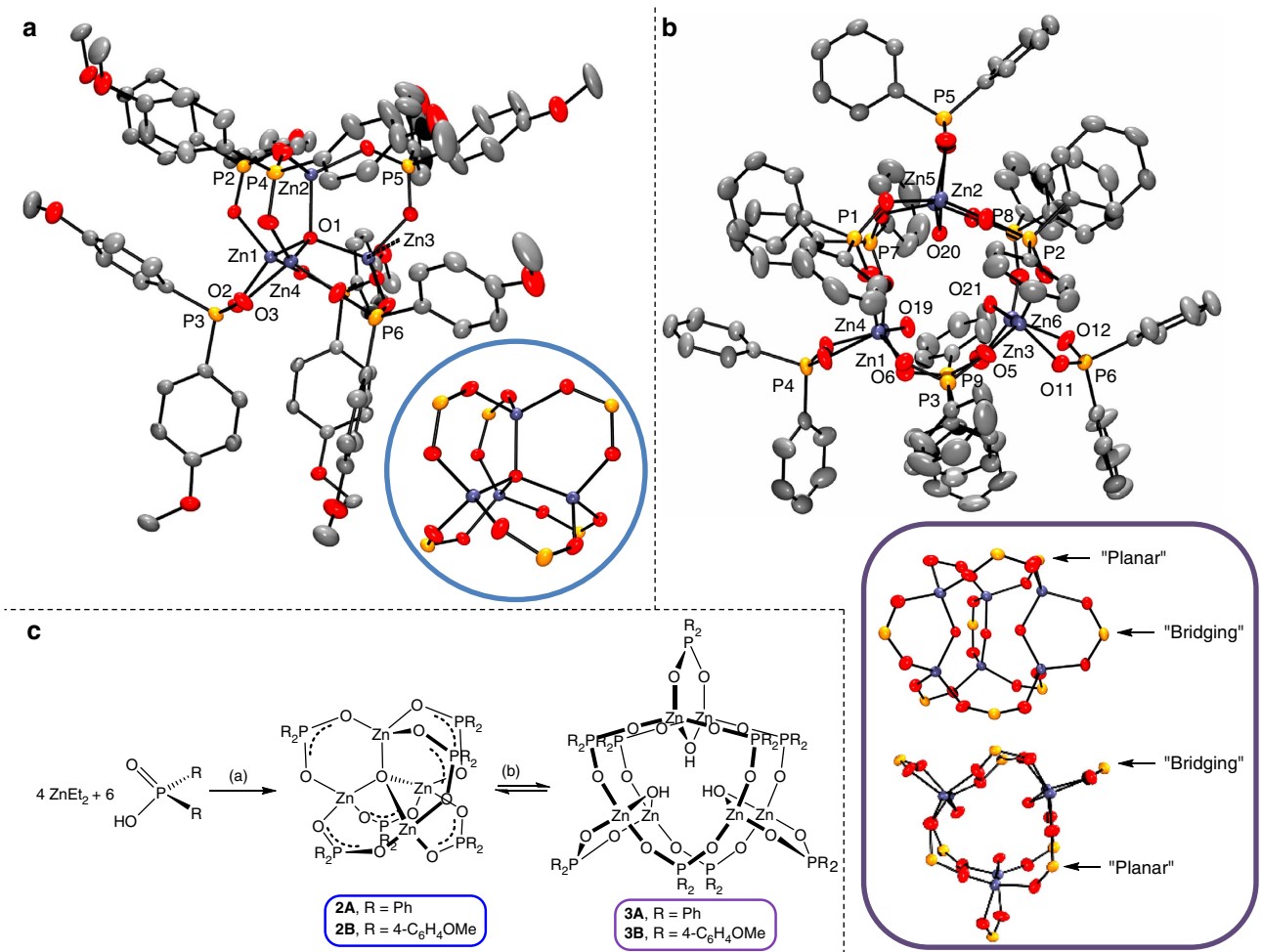

**Figure 3 | Synthetic path and solid-state structures of 2B and 3A.** Structures: (**a**) **2B** and (**b**) **3A** (H-atoms omitted for clarity). Views of the Zn cluster core structures, with aromatic groups omitted, are provided inside the coloured circle/box c) Synthesis and equilibrium of **2A/B** and **3A/B**. Reagents:(**a**) 1 eq. $H_2O$, toluene or $CH_2Cl_2$. (**b**) 5 eq. $H_2O$.

for stabilization[46,49]. Zinc-hydroxy species are of interest in a range of contexts, including as putative intermediates during ZnO nanoparticle synthesis and as models for a range of zinc-dependent metalloenzymes[11,45–47,50–54].

Given interest in similar carboxylate-ligated Zn clusters, the different structures and reactivity observed here with phosphinate ligands is notable. The solid-state structure of **2B** shows $Zn-$phosphinate bond lengths ranging from 1.917(2) to 1.960(2) Å, with an average (1.936(2) Å) slightly greater than that of the analogous Zn-benzoate structure $Zn_4O(O_2CPh)_6$ (average = 1.926 Å)[35], in line with slightly weaker bonding from the phosphinate. The $Zn-(\mu_4\text{-}O)$ bonds in **2B** are also lengthened (average **2B**, 1.989(2) Å; $Zn_4O(O_2CPh)_6$ 1.946 Å), presumably as a result of the larger size of the phosphinate chelate compared with a carboxylate (average $P-O$ (**2B**), 1.512(2) Å; average $C-O$ $(Zn_4O(O_2CPh)_6)$, 1.258(2) Å)[35], which allows for an expansion of the $Zn_4O$ cluster. Compounds **2A/B** react with water to form well-defined zinc hydroxide complexes, whereas the carboxyate analogue $[Zn_4O(CO_2Ph)_6]$ reacts as a Lewis acid towards water to form an aqua complex $[Zn_4(\mu_4\text{-}O)(OOCPh)_6(H_2O)(THF)]$[31]. Thus, complexes with phosphinate ligands undergo disruption of the $Zn_4O$ core. The $Zn-OH$ bonds in **3A** are shorter on average (1.935(2) Å) than the $Zn-(\mu_4\text{-}O)$ bonds in **2B** (1.989(2) Å); it may be that owing to the larger size of the phosphinate ligand, effective bonding to the oxo/hydroxo ligand is favoured in the expanded $Zn_6$ structure.

**Equilibrium studies**. To explore the factors controlling the equilibration of zinc-oxo and zinc-hydroxy clusters, variable-temperature NMR spectroscopy was applied, using a solution containing a starting 2:3 ratio of **2A:3A**, over the temperature range 288–328 K (Supplementary Fig. 24). At each temperature, the equilibrium was rapidly established, as confirmed by an identical second spectrum, obtained after ∼15 min. The ratio of **2A:3A** is easily determined from the $^{31}P\{^1H\}$ NMR spectra (see Supplementary Methods), with **2A** being the major species at temperatures above 318 K. Under the experimental conditions, the concentration of water is low (0.059 M) but is all fully dissolved with no downfield signal, which would be expected from separated water droplets. Van't Hoff analysis showed that $\Delta H_r = -108 \pm 3$ kJ mol$^{-1}$ and $\Delta S_r = -238 \pm 9$ J K$^{-1}$ mol$^{-1}$ (Supplementary Fig. 25 and Supplementary Methods). Clearly, the hydroxo structure **3A** is enthalpically favoured, but the entropic advantage results in **2A** becoming dominant at higher temperatures. A similar equilibrium exists between zinc-oxo cluster **2B**, water and **3B** (Supplementary Figs 22 and 23). The equilibrium lies more towards the zinc oxo species, **2B**, than in the **2A/3A** system (**2B/3B** = 1:0.23 *cf.* **2A:2B** = 1:0.9, 2.3 eq. water added). Van't Hoff analysis revealed that $\Delta H_r = -97 \pm 3$ kJ mol$^{-1}$ and $\Delta S_r = -234 \pm 9$ J K$^{-1}$ mol$^{-1}$ (Supplementary Fig. 26 and Supplementary Table 1). Compared with **2A/3A**, the entropy of reaction is unchanged (within error), but the zinc hydroxyl cluster, **3B**, is slightly less enthalpically favoured

(Supplementary Table 1). These results provide a thermodynamic rationale for the equilibration between the clusters and demonstrate the importance of the phosphinate ligand in controlling the relative stabilities of the clusters.

**Synthesis of a zinc-boroxine cluster.** The proximity of the three hydroxyl groups in **3A** suggests the intriguing possibility of coordination of further atoms/molecules in the centre of the cluster (O − centroid distances 1.5–1.8 Å, Supplementary Fig. 21). In a different system and geometry, partially condensed trisilanol silsequioxanes have been widely used to bind heteroatoms for catalytic and other studies[55]. The reactivity of **3A** with organometallic reagents (such as AlEt$_3$) is challenging, especially given the presence of water in the solution equilibrium, which results in preferential hydrolysis of the organometallic species, driving the equilibrium back towards **2A**. An alternative approach is to use a different oxygen source to form the Zn–O–X moieties. In this regard, boric acid (B(OH)$_3$) is attractive for its aqueous stability and trigonal planar shape. Boric acid clearly reacts with a THF solution of **2A/3A**, leading to the formation of a product **4A** (Supplementary Figs 27–29). Compound **4A** can also be prepared in quantitative yield ($^{31}$P NMR) by the direct reaction of a 2:3:1 ratio of ZnEt$_2$, DPPA-H and boric acid, in THF (Fig. 4). Again, an equilibrium exists between **4A**, **2A** and **3A** (Supplementary Fig. 33); when 17 eq. of water was added to a solution of pure **4A**, a molar ratio of 89:7:4 for **4A:2A:3A** formed,

showing that **4A** is favoured even under wet conditions. Crystals of **4A**, grown from THF/hexane, showed the structure as [Zn$_6$B$_3$O$_3$(DPPA)$_9$] (Fig. 4b and Supplementary Figs 30 and 32). The planar cluster contains six zinc atoms surrounding a B$_3$O$_3$ core. Each zinc atom is tetrahedrally coordinated to three bridging phosphinate ligands and a μ$_3$-oxo ligand. The oxo ligands are each also coordinated to the boroxine core. Two phenyl substituents align above and below this boroxine core, suggesting some π–π stacking exists in the solid state, it is well known that boroxines exhibit partial aromaticity[56]. The structure of **4A** is quite different to that of **2A** or **3A** and it is proposed that the spontaneous self-assembly is driven by the planar boroxine core. The Zn$_6$B$_3$O$_3$ cluster planarity may also be relevant for the construction of more complex two-dimensional materials, including MOFs. The structure of **4A** is maintained in solution; two singlet signals in the $^{31}$P NMR spectrum are observed in a 2:1 ratio (22.7, 29.3 p.p.m.) as expected from the two environments (in and out of the plane) in the solid-state structure (Supplementary Fig. 27). The $^1$H NMR spectrum shows three sets of phenyl resonances in a 1:1:1 ratio (Supplementary Fig. 28).

**Synthesis of a partially hydrolysed Zn$_{11}$ cluster.** It is of interest to consider what role clusters such as **1–3** might take during the formation of phosphinate-coordinated zinc oxide nanoparticles by hydrolysis routes. We have previously reported the potential to introduce sub-stoichiometric quantities of

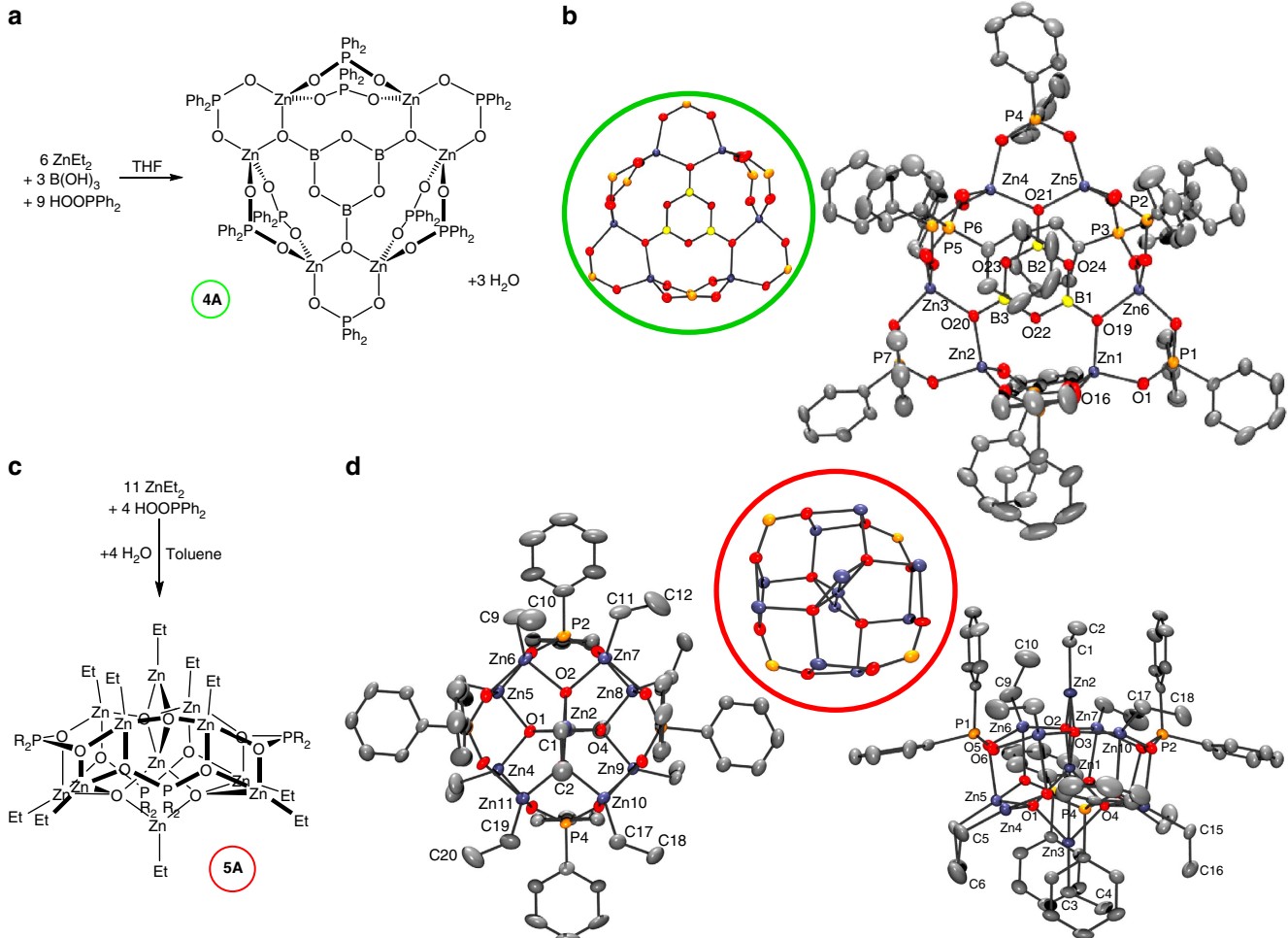

**Figure 4 | Synthetic path and solid-state structures of 4A and 5A.** Schemes showing synthesis of (**a**) **4A** and (**c**) **5A**. Solid-state structures of (**b**) **4A** and (**d**) **5A** (2 views shown) (views of the Zn cluster core structures, with the phenyl/ethyl groups omitted, are provided inside the coloured circles).

carboxylic acid/phosphinic acid during $ZnEt_2$ hydrolysis, to deliver surface-ligated crystalline ZnO nanoparticles with well-defined sizes (2–4 nm). The capped nanoparticles show good solubility in organic solvents and have been used as quasi-homogeneous catalysts as well as in the preparation of high-loading fraction ZnO-polymer composites[12,20]. In general, there is significant interest in the preparation of ZnO nanoparticles by the controlled hydrolysis of organozinc reagents, including $ZnEt_2$, as it provides a room-temperature method to crystalline nanoparticles and a route to useful inorganic hybrid materials[10,12,16,57]. So far, however, the mechanism and intermediates implicated in the hydrolysis of well-defined organometallic reagents, with or without capping ligands, to nanoparticles is not at all well understood[19]. As a starting point to understanding how the particles form, we proposed that there may be some partly hydrolysed clusters present. The hydrolysis reaction occurs in solutions, often of inert organic solvents; thus, it is beneficial to apply solution-based spectroscopic techniques. A particular benefit of phosphinate ligands, as noted above, is the facility to apply $^{31}P\{^1H\}$ NMR spectroscopy. Previous studies of ZnO nanoparticles have shown they approach surface saturation with ligand, when a mixture of 5 eq. of $ZnEt_2$ with one equivalent of ligand (typically dioctylphosphinic acid) is hydrolysed[20]. Introducing the water gradually allows the speciation during this process to be probed. Using DPPA as a model ligand and adding only one equivalent of water to this 5:1 mixture, a new phosphorus-containing cluster compound was identified by NMR spectroscopy (Supplementary Figs 34–36). By adjusting the ratios to favour this new species, we were able to form crystals from an 11:4:4 mixture of $ZnEt_2$, $H_2O$ and DPPA-H. The isolated crystals revealed a cluster containing 11 zinc atoms, $[Zn_{11}Et_{10}O_4(DPPA)_4]$; elemental analysis was also in good agreement (Fig. 4d and Supplementary Fig. 37). Compound **5A**, $[Zn_{11}Et_{10}O_4(DPPA)_4]$, can be thought of as an extension of **1A** in which 6 extra Zn−Et groups are added along with a central $ZnO_4$ tetrahedron. Unlike **1A**, the bonding within the phosphinate ligand is now delocalized with equivalent P−O bonds throughout. Compound **5A** has approximate $D_{2d}$ point symmetry, with eight Zn−Et groups coordinated by bridging phosphinate ligands surrounding a central $ZnO_4$ tetrahedron. A further two Zn−Et groups are located above and below the central $ZnO_4$ core, without any bonds to phosphinate ligands; these two zinc atoms are three coordinate (trigonal planar). The phosphinate–Zn bonds are somewhat variable (1.870(2)–2.094(2) Å; *cf.* **2B**, 1.917(2)–1.960(2) Å), suggesting the central core dictates the geometry. In solution, the $^1H$ NMR spectrum indicates a similar structure, with two different zinc-coordinated ethyl environments in a 4:1 ratio (Supplementary Fig. 35). The two ethyl ligands at the three coordinate zinc centres are significantly shifted (−1.48, 0.23 p.p.m.) presumably due to proximity to electron-deficient zinc centres (Supplementary Fig. 36). The other zinc ethyl ligands show diastereotopic methylene proton signals, due to chirality at those zinc centres.

To understand the cluster interconversions, 15 eq. of $ZnEt_2$ were added to 4 eq. of **2A**, leading to a 5:1 ratio of **1A:5A**, together with residual **2A** and $ZnEt_2$. Increasing the temperature drives the backward reaction and increases the relative proportions of **2A** and $ZnEt_2$, indicating that an equilibrium exists between the four species (Fig. 5 and Supplementary Figs 38–43). Establishing the same equilibrium from **2B** also revealed a 5:1 ratio of new clusters (that is, the analogues **1B** and **5B**; Supplementary Figs 40 and 41, and Supplementary Table 2), highlighting the generality of these reactions with different phosphinate ligands. Furthermore, all the room-temperature $^1H$ NMR spectra of **3A**, **3B** and **4A** show broadening of the phenyl environments for the phosphinates in asymmetrical

environments, indicating that ligand rotation allows exchange between phenyl environments. This rotation may suggest that the flexible ligand coordination of phosphinate ligands enable rearrangements to the thermodynamic products.

**In situ identification of clusters in nanoparticle synthesis.** Having established the structures of clusters **1–5A** and the generality to other related model ligands, their roles in ZnO nanoparticle synthesis was explored. In particular, the hydrolysis of $ZnEt_2$ was performed in the presence of the di-octylphosphinate (DOPA) ligands, which are representative of systems used to sterically stabilize nanoparticles. The long alkyl chains hinder crystallization, but the utility of the $^{31}P$ NMR handle allows characterization. Using analogous procedures to the preparation of **1A/B** and **2A/B**, the clusters **1C** $(Zn_4Et_4(DOPA)_4)$ and **2C** $(Zn_4O(DOPA)_6)$ were easily identified following the reactions of appropriate ratios of $ZnEt_2$ with DOPA-H (and water for **2C**) (Supplementary Figs 44–48). Addition of an excess of water ($\sim6$ eq.) to **2C** leads to two $^{31}P$ signals (which sum to 1% of the total integral considering residual **2C**) indicative of the formation of **3C** $(Zn_6(OH)_3(DOPA)_9)$ (Supplementary Fig. 49). However, in this case, the equilibrium strongly favours **2C**, likely to be due to the greater steric hindrance from the bulky ligand. Nevertheless, the speciation using the long-chain phosphinate maps onto the model clusters already structurally characterized.

For the synthesis of functionalized nanoparticles, a Young's tap NMR tube was loaded with a 5:1 ratio of $ZnEt_2$ and DOPA-H solvated in $d_8$-toluene, similar to conditions established previously[20]. Water was added in sequential 0.5 μl aliquots to the NMR tube (each aliquot making up 12.5% of the 4 μl required for total hydrolysis) under a flow of $N_2$. NMR spectra were recorded at each point along the hydrolysis pathway (after $\sim30$ min reaction time at each point, Supplementary Figs 50–54). An internal standard ($PPh_3$ in a capillary tube) was used to allow calculation of relative integrals from the $^{31}P\{^1H\}$ NMR spectra and to monitor the evolution of the cluster species during the reaction (Fig. 6b,c). It is worth noting that the reactions described above were performed at stoichiometries targeted to particular clusters, whereas the reactions on the path to nanoparticles occur in the presence of excess diethyl zinc. Initially, in the absence of water, **1C** was observed alongside the excess $ZnEt_2$, as expected. Initial hydrolysis of the remaining Zn−Et bonds results in the loss of signal for **1C** and the formation of a signal at 61.0 p.p.m., assigned to complex **5C** $[Zn_{11}Et_{10}O_4(DOPA)_4]$ (see Supplementary Note 1), accompanied by various low-intensity new signals. Compound **5C** can be independently synthesized and also forms an equilibrium with **1C**, **2C** and $ZnEt_2$ as shown in Fig. 5 (Supplementary Figs 55–58). The other minor species (shown in Fig. 6c) are currently not assigned; the strongest signal is tracked in Fig. 6b and disappears after 38% total hydrolysis, leaving **5C** as the dominant ligated species until hydrolysis nears completion (Fig. 6). In the accompanying series of $^1H$ NMR spectra (Supplementary Fig. 51), the sharp ethyl signals for **5C** also grow in and persist almost until hydrolysis is completed. The initially exchange-broadened signals for free $ZnEt_2$ sharpen as **1C** and other clusters are consumed and then disappear as hydrolysis proceeds. Although the NMR spectra change little between 38 and 63% hydrolysis, a distinctive yellow colour emerges. It appears that the extra water reacts with $ZnEt_2$ to produce species that are silent in the $^1H$ and $^{31}P$ NMR spectra. This same yellow colour is observed during hydrolysis of $ZnEt_2$ in the absence of ligands and has been noted by other researchers preparing ZnO nanoparticles, although the speciation remains

**Figure 5 | Cluster equilibrium.** Equilibrium between **2**, **1** and **5**. Complexes **1B**, **5B**, **1C** and **5C** identified by NMR spectroscopy only.

unexplored[57]. Transmission electron microscopy (TEM) and powder XRD of this 'yellow ZnO' derived from a reaction of $ZnEt_2$ with 0.75 eq. of $H_2O$ in toluene revealed a mainly amorphous polydispersed and agglomerated nanoparticulate material displaying a very broad, weak ZnO diffraction pattern (Fig. 6d and Supplementary Figs 59–61). Elemental analysis confirmed that some ethyl groups were also maintained, although no sharp signals attributable to ethyl moieties were observed in the $^1H$ NMR spectra. To monitor the formation of crystalline ZnO, the hydrolysis reaction, in the presence of DOPA ligands, was also followed by ultraviolet spectroscopy (290–400 nm; Supplementary Fig. 62). No optical absorption was observed in either the starting molecular precursors or 25% hydrolysed mixtures, confirming that the yellow colour is not associated with ligated species, **1C** or **5C**, but rather the product of partial $ZnEt_2$ hydrolysis, as noted above[57]. At 50% hydrolysis, a strong absorption is observed, which reduces smoothly, from 290–350 nm, but with no visible band edge typically associated with a crystalline ZnO species; this absorption is similar but stronger at 75% hydrolysis, as consumption of $ZnEt_2$ continues, and is attributed to highly defective/disordered amorphous ZnO nanoparticles, as discussed above[58]. However, at 100% hydrolysis, the ultraviolet absorption drops significantly (consistent with the observed loss of yellow colour), giving a typical ZnO band edge signal, corresponding to nanoparticles with a diameter of $\sim3$ nm (Supplementary Fig 63)[59].

The final product of hydrolysis is colourless ZnO nanoparticles (2–3 nm) capped by di(octylphosphinate) ligands (ZnO@DOPA), extensively characterized previously (and here shown in Fig. 6e and Supplementary Figs 64–72), showing well-isolated crystalline ZnO cores in TEM (Supplementary Fig. 64), clear XRD features of Wurtzite ZnO (Supplementary Fig. 69), an identifiable organic content in elemental analysis, and maintaining a high solubility in toluene (unlike the hydrolysis product in the absence of phosphinate)[20]. Both NMR and ultraviolet spectroscopy studies indicate that the ligand-capped ZnO@DOPA nanoparticles only form towards the end of the hydrolysis reaction (Fig. 6 and Supplementary Fig. 62). The nanoparticles are observed by $^{31}P$ NMR spectroscopy as a broad signal after 75% hydrolysis (52 p.p.m. (full width at half maximum $\sim1,800$ Hz)); the signal breadth is typical for ligands coordinated to nanoparticle surfaces (Supplementary Figs 66 and 67)[17]. After all the Zn − Et bonds are hydrolysed, a second new species (**2C**) is observed along with the nanoparticles in the $^{31}P$ NMR spectra (Fig. 6b,c). Although the majority of phosphinate is bound to nanoparticles, the zinc oxo cluster **2C** forms as a significant byproduct (25% of total ligand). The presence of **2C**, which has a high DOPA:Zn ratio, is consistent with the liberation of DOPA on the final hydrolysis of **5C** and agglomeration of ZnO units (note: **3C** was not observed

here as it only forms as a very minor equilibrium partner with **2C** in the presence of excess moisture). Compound **2C** is soluble in acetone, unlike ZnO@DOPA nanoparticles, and thus can be easily removed, allowing the isolation of pure nanoparticles. It is notable also that Chaudret and colleagues[16], and Mayer and colleagues[7] have recently both reported that amine-ligated zinc oxide surfaces also exhibit exchange between coordinated and 'free' ligands; it is interesting to consider whether well-defined molecular zinc cluster complexes may also be present in these cases. However, low temperature and diffusion-ordered NMR spectroscopy studies (Supplementary Figs 66–68) showed no evidence of ligand exchange between the nanoparticles and **2C**.

This reaction trajectory is quite unexpected. The highly moisture sensitive alkyl–zinc complex **5C** forms rapidly and is maintained throughout the majority of the hydrolysis reaction, sequestering essentially all the available ligand, whereas residual $ZnEt_2$ is consumed to form unligated ZnO nanoparticle precursors. A 50% hydrolysed mixture was monitored and found to be unchanged after 15 h, indicating that the system is in thermodynamic equilibrium, with 93% of the total phosphinate supply incorporated in the form of a stable cluster (minor unidentified species make up the balance to $\sim100\%$ relative to the internal standard; Fig. 6c and Supplementary Figs 52 and 53). It is only near-full hydrolysis of all other Zn − Et species that **5C** reacts with the 'yellow' ZnO precursors to form the phosphinate capped ZnO nanoparticles. Unlike previous reports in the literature, proposing cluster compounds as molecular building blocks that directly map onto the final nanoparticle (NP) crystal structure[60], here the cluster compounds do not obviously relate to Wurtzite and instead appear to act only as a reservoir of ligand. This fresh insight into ligand behaviour during nanoparticle synthesis has implications for the concepts of nanoparticle growth and stabilization.

The formation of ZnO nanoparticles by hydrolysis, under the same conditions, but in the absence of any ligand produces insoluble nanoparticles, with average particle sizes of $\sim3.5$ nm (by XRD; Supplementary Fig. 73). The similarity in size range to the particles prepared using the DOPA ligands (2–3 nm by XRD) indicates that the ligand is not critical for particle size control, in keeping with the distribution of ligands only at the end of the reaction. Previous studies using carboxylate ligands and a similar synthetic protocol also found that ZnO particles were consistently formed within the 3–5 nm size range, regardless of the nature or loading of the ligand applied[12]. Nonetheless, the ligands are important to produce well-dispersed and soluble nanoparticles, as they prevent aggregation observed in their absence. Furthermore, ligands are likely to have a significant impact on subsequent ripening and ageing of the nanoparticles[12,19].

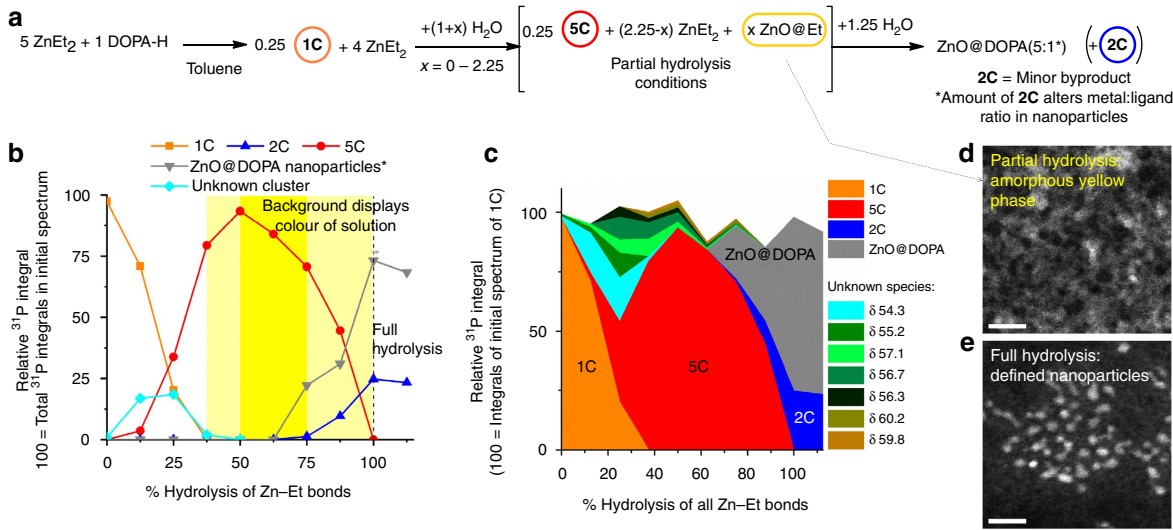

**Figure 6 | In situ study of ZnO nanoparticle synthesis.** (**a**) Scheme showing the synthesis of ZnO nanoparticles. (**b**) Relative integrals and (**c**) area graph showing sum of integrals from $^{31}P\{^{1}H\}$ NMR spectra of the P-containing species, formed on increasing amounts of added water to an original 5:1 $ZnEt_2$: DOPA-H mixture (*the signal for the ZnO@DOPA nanoparticles is broad, leading to less accurate integration). Alongside the unknown cluster displayed in **b**, small quantities of other unknown species (always <10% total integral) were also observed between 0–82% hydrolysis (shown in c and labelled by their $^{31}P$ NMR signal (Supplementary Figs 50–53 for spectra and Supplementary Fig. 54 for %Zn speciation plots)). Area graph (**c**) shows that the sum of relative integrals remains close to 100% of the initial spectrum (**1C** only species) throughout the reaction. (**d,e**) Representative scanning TEM (STEM) images (in annular dark field mode) of the partially hydrolysed $ZnEt_2$ (ZnO@Et yellow phase) and fully hydrolysed ZnO@DOPA nanoparticles at the same level of magnification (scale bar, 10 nm; note, Zn-containing phase appears pale on a dark background).

## Discussion

This study exploits an organometallic route to nanoparticles that delivers only the stoichiometric quantity of ligand. By avoiding the excess uncoordinated ligand used in many liquid-phase nanoparticle syntheses, the fate of the ligand at various stages of the reaction can be directly determined. Nanoparticle nucleation is very often considered to be a non-equilibrium process, requiring high degrees of super-saturation and high concentrations of active surfactants to minimize the size of the critical nuclei, with particle size often controlled by kinetics, requiring hot injection, fast mixing and the like[61]. Alternatively, sol-gel approaches often involve irreversible condensation reactions[20]. The presence of ligands during nanoparticle synthesis is usually assumed to reduce the nucleation barrier and critical nucleus size, by reducing surface energy of the nascent nanoparticle. Smaller particles can therefore form, which are sterically stabilized against coalescence by the coordinated ligand[62–64]. Here, this model is completely subverted as the ligands are observed to only interact at the end of the reaction. The behaviour found in this ZnO system may well be observed in other cases, where ligand supply is limited[65,66], or in systems with analogous structures, such as (Zn, Cd) (S, Se, Te) capped by coordinating ligands (carboxylates, phosphinates or phosphonates).

This study shows that equilibrium cluster interconversions, including of oxo-bridged species, can play a key role in the distribution of ligand on growing nanoparticles. Understanding the mechanisms by which the nanoparticle core is formed and then decorated by ligands is likely to help in the formation of (surface) doped nanoparticles crucial for many applications[1,67,68], especially in (opto)electronics, and in forming mixed ligand layers, which may allow unusual wettability or adaptive behaviour[69,70].

In conclusion, the reactions between diethyl zinc, phosphinic acids and water lead to a rich variety of new clusters. Using equimolar diethyl zinc and phosphinic acid yields an organometallic cubic structure, **1**; the species is hydrolysed, by water to a $Zn_4O$ cluster, **2**. The $Zn_4O$ cluster equilibrates with excess water to produce hexa-zinc tris(hydroxide) trigonal prismatic complexes, **3**.

The equilibrium is very unusual, yet provides a simple means to prepare zinc hydroxide clusters; such species are usually significantly more challenging to prepare, yet are important structures in bio-inorganic and other processes. A new planar $Zn_6B_3$ cluster, **4A**, is formed with $B(OH)_3$ taking the role of the water in the 'hydrolysis' of the Zn–Et bonds. Finally, **5A**, a cluster containing 11 zinc atoms, was prepared by partial hydrolysis of Zn–Et bonds. Its analogue, coordinated by a long-chain di(octyl phosphinate) ligand, **5C** plays a crucial role in the synthesis of ZnO@DOPA nanoparticles. Interestingly the reactive cluster is retained as a spectator after its formation during the initial hydrolysis, sequestering all the available DOPA ligand in a stable form, leaving water to react directly with residual unligated ethyl-zinc species. Only on approaching total hydrolysis of all zinc alkyl functionalities is the ligand delivered to the growing nanoparticle surface. The final re-equilibration step converts disordered polydispersed nanoparticle precursors into the well-defined ligand-capped ZnO product.

In addition to the rich cluster chemistry identified, using simple, commercial reagents, relevant as single-source precursors to prepare (optionally doped) nanomaterials or as nodes in new families of metal-organic frameworks, this study highlights a number of useful principles. First, phosphinates are under-utilized ligands, which provide a diagnostic $^{31}P$ NMR handle; a strong and distinct NMR signal from the binding group is extremely helpful for navigating complex mixtures to identify the correct stoichiometries for pure products. Second, the cluster species form through a series of reversible equilibria as hydrolysis proceeds, but specific products can be isolated directly once their composition is identified. These equilibrium processes should allow further investigation and interpretation of the nucleation and growth of nanoparticles.

## Methods

**Experimental details.** All manipulations were undertaken using a nitrogen filled glovebox or using a Schlenk line, unless otherwise stated. DPPA and bis(4-methoxyphenyl) phosphinic acid were used directly from suppliers and

di-octylphosphinic acid was prepared using an established literature route[71]. ZnO@DOPA nanoparticles were prepared using a literature route[20], and synthetic details and characterization of ZnO@DOPA, ZnO and 'yellow ZnO' are included in the Supplementary Methods. $ZnEt_2$ is pyrophoric (caution!) and was added to samples in a nitrogen-filled glovebox. As a liquid, all additions of $ZnEt_2$ were transferred by syringe (measured by negative weight of donor flask). THF was dried by refluxing over sodium and benzophenone, and stored under nitrogen. Hexane and toluene were pre-dried over potassium hydroxide and then further dried by refluxing over sodium (benzophenone, for hexane) and stored under nitrogen. 'Extra-dry' acetone was purchased from Acros Organics. All dry solvents and reagents were degassed by three freeze–pump–thaw cycles and stored under nitrogen. Solvents were tested for moisture content by Karl Fischer Titration (Mettler Toledo): toluene, 3.8 p.p.m.; THF, 4.1 p.p.m.; dichloromethane, 1.7 p.p.m.

NMR spectra were recorded on Bruker AV-400 or AV-500 instruments and all chemical shifts reported in p.p.m. Solid-state Fourier transform infrared spectra were recorded using a Perkin-Elmer Spectrum 100 FT-IR spectrometer with a Universal ATR Sampling Accessory. Ultraviolet spectroscopy was recorded using a PerkinElmer Lambda 950 spectrophotometer, from toluene solutions. All mass spectrometry measurements were performed using a MALDI micro MX micromass instrument. Isotope patterns were compared with predicted patterns using mMass. Elemental Analysis was determined by Stephen Boyer at London Metropolitan University. Thermo-gravimetric analysis was undertaken under an air atmosphere, using a Mettler/Toledo TGA/DSC 1LF/UMX instrument at a heating rate of $10\,K\,min^{-1}$. Powder XRD was performed using an X'Pert Pro diffractometer (PANalytical B. V., The Netherlands) and X'Pert Data Collector software, version 2.2b. The instrument was used in the theta/theta reflection mode, fitted with a nickel filter, 0.04 rad Soller slit, 10 mm mask, 1/4° fixed divergence slit and 1/2° fixed antiscatter slit. The diffraction patterns were analysed using Fityk (version 0.9.0; Marcin Wojdyr, 2010), the peaks were fitted to a SplitPearson7 function and the particle size was calculated using the fitted full-width half-maximum using the Scherrer equation. Scanning TEM images, conventional TEM images and electron diffraction patterns were acquired on an FEI Titan 80–300 microscope operated at 300 kV. For air-sensitive samples, the sample solution was deposited on a 400-mesh copper holey carbon grid with an ultra-thin 3 nm-thick carbon support (Agar Scientific AGS187-4), while in a glove box. The grid was then loaded into a Gatan environmental holder to prevent any exposure to air prior to TEM imaging. Further details of Van't Hoff analysis, NMR equilibrium studies, in situ experiments and single crystal XRD are included in the Supplementary Methods and supplementary Table 3.

**Syntheses and characterization of 1A.** $Zn_4Et_4(DPPA)_4$: DPPA-H (88.3 mg, 0.405 mmol) was placed in a Young's tap flask and dissolved in $CH_2Cl_2$ ($\sim 3$ ml). To this, $ZnEt_2$ (50 mg, 0.405 mmol) was added and the evolution of ethane gas was observed. Hexane was layered onto the solution, allowing the growth of white crystalline **1A** over several days (Isolated yield: 35 mg, 23%). Alternatively, toluene can be used as the reaction solvent and **1A** precipitates out directly as a powder in this case (42% yield). Compound **1A** is highly moisture sensitive and traces of **2A** can be observed in its NMR spectra if there is any contamination by trace moisture. $^{31}P\{^1H\}$ **NMR** (162 MHz, $CDCl_3$): $\delta$ 23.2 (s, 4P) p.p.m.; $^1H$ **NMR** (400 MHz, $CDCl_3$): $\delta$ 0.41 (q, $CH_2$, $J_{HH} = 8$ Hz, 8H), 1.33 (t, $CH_3$, $J_{HH} = 8$ Hz, 12H), 7.08 (td, DPPA, $J_{HH} = 8$ Hz, 3 Hz, 16H), 7.26 (m, DPPA, 8H), 7.34 (m, DPPA, 16H); anal. calcd for: $Zn_4P_4O_8C_{56}H_{60} = C$, 53.96; H, 4.85; found C, 53.82; H, 4.76%.

**Syntheses and characterization of 1C.** $Zn_4Et_4(DOPA)_4$: 47 mg (0.162 mmol) dioctylphosphinic acid was placed in a Young's cap NMR tube and dissolved in $CDCl_3$ (0.5 ml). To this, 20 mg (0.162 mmol) of $ZnEt_2$ was added and the evolution of ethane gas was observed. The product was analysed by NMR spectroscopy directly and **1C** was identified as $\sim 68\%$ of $^{31}P$ NMR signal with the remainder as broad unidentified products. If an excess of $ZnEt_2$ is present, **1C** forms as the sole $^{31}P$-containing species. $^{31}P\{^1H\}$ **NMR** (162 MHz, $CDCl_3$): $\delta$ 50.9 (s, 4P); $^1H$ **NMR** (400 MHz, $CDCl_3$): $\delta$ 0.03 (q, Et $CH_2$, $J_{HH} = 8$ Hz, 8H), 0.91 (t, DOPA $CH_3$, $J_{HH} = 7$ Hz, 24H), 1.14 (t, Et $CH_3$, $J_{HH} = 8$ Hz, 12H), 1.30 (br, DOPA $CH_2$, 80H), 1.53 (br, DOPA $\beta$-$CH_2$, 16H), 1.68 (m, DOPA $\alpha$-$CH_2$, 16H) (400 MHz, $d_8$-toluene): $\delta$ 0.62 (q, Et $CH_2$, $J_{HH} = 8$ Hz, 8H), 0.91 (t, DOPA $CH_3$, $J_{HH} = 7$ Hz, 24H), 1.31 (br, DOPA $CH_2$, 80H), 1.66 (t, Et $CH_3$, $J_{HH} = 8$ Hz, 12H), 1.8 (br, DOPA $\beta$-$CH_2$, 16H), 1.9 (m, DOPA $\alpha$-$CH_2$, 16H)

**Syntheses and characterization of 2A.** $Zn_4(\mu_4$-O$)(DPPA)_6$: DPPA-H (265 mg, 1.21 mmol) was placed in a Schlenk flask with a stirrer bar and suspended in toluene ($\sim 15$ ml). To this, $ZnEt_2$ (100 mg, 0.81 mmol) was added and the evolution of ethane gas was observed. The solution was stirred for 30 min, before addition of a 3.6 µl (0.2 mmol) of water by Eppendorf pipette, under a flow of nitrogen, and the mixture stirred overnight. A precipitate formed from the reaction solution; gentle heating allowed re-solvation of this material and then the flask was allowed to stand at room temperature, to allow the formation of a colourless crystalline material (isolated yield = 264 mg, 83%). Alternatively, **2A** can be synthesized using $CH_2Cl_2$ as the reaction solvent and crystals can then be formed by layering the solution with hexane. It is notable that if there is any excess of water then traces of **3A** are observable. $^{31}P\{^1H\}$ **NMR** (162 MHz, $CDCl_3$): $\delta$ 32.9

(s, 6P) p.p.m.; $^1H$ **NMR** (400 MHz, $CDCl_3$): $\delta$ 7.08 (td, DPPA, $J_{HH} = 8$ Hz and 3 Hz, 24H), 7.31 (m, DPPA, 12H), 7.51 (m, DPPA, 24H) p.p.m.; m/z (MALDI–ToF, matrix = 9-nitroanthracene, solvent = $CHCl_3$): 1363.2 $\{[Zn_4O(DPPA)_5]^+$ calcd 1362.9}, 1619.3 $\{[Zn_4O(DPPA)_6.K]^+$ calcd 1618.9} amu; anal calcd for $Zn_4P_6O_{13}C_{72}H_{60} = C$, 54.71; H, 3.83%; found C, 54.33; H, 4.23% (note: traces of $CH_2Cl_2$ and hexane were observed by NMR spectroscopy when these crystals were solvated even after thorough drying under vacuum).

**Syntheses and characterization of 2B.** $Zn_4(\mu_4$-O$)(D^{MeO}PPA)_6$: Bis(4-methoxyphenyl)-phosphinic acid (337.9 mg, 1.21 mmol) was placed in a Schlenk flask with a stirrer bar and dissolved in $CH_2Cl_2$ (20 ml). To this, $ZnEt_2$ (100 mg, 0.81 mmol) was added and the evolution of ethane gas was observed. The solution was stirred for 30 min, before addition of a solution of water (4 µl, 0.22 mmol) in dry acetone (0.1 ml). The volume of solvent was reduced to $\sim 7$ ml, by vacuum, before layering the solution with hexane ($\sim 40$ ml), to allow the formation of crystals (isolated yield = 210 mg, 80%). Alternatively, **2B** was synthesized in toluene solvent and crystals formed directly from the reaction medium. This product, which crystallizes with two molecules of toluene, was rather insoluble once formed. $^{31}P\{^1H\}$ **NMR** (162 MHz, $CDCl_3$): $\delta$ 32.8 (s, 6P) p.p.m.; $^1H$ **NMR** (400 MHz, $CDCl_3$): $\delta$ 3.75 (s, OMe, 36H), 6.58 (m, $C_6H_4$OMe, 24H), 7.47 (m, $C_6H_4$OMe, 24H) p.p.m.; m/z (MALDI–ToF, matrix = 9-nitroanthracene, solvent = $CHCl_3$): 1663.4, $\{[Zn_4O(D^{MeO}PPA)_5]^+$ calcd 1663.0} amu.; anal. calcd for $Zn_4P_6O_{25}C_{84}H_{84} = C$, 51.98; H, 4.36%; found C, 51.84; H, 4.43%; anal. calcd for (product isolated from toluene) $Zn_4P_6O_{25}C_{84}H_{84}.2(C_7H_8) = C$, 55.39; H, 4.74%; found C, 55.24; H, 4.98%.

**Syntheses and characterization of 2C.** $Zn_4(\mu_4$-O$)(DOPA)_6$: (529 mg, 1.82 mmol) dioctylphosphinic acid was placed in a Schlenk flask with a stirrer bar and dissolved in toluene ($\sim 12$ ml). To this, 150 mg (1.21 mmol) of $ZnEt_2$ was added and the evolution of ethane gas was observed. The solution was stirred for 30 min before addition of a solution of water in acetone (6 µl (0.33 mmol) water in 0.2 ml acetone). The solvent may be removed to leave an oily colourless product. $^{31}P\{^1H\}$ **NMR** (162 MHz, $CDCl_3$): $\delta$ 56.5 (s, 6P); $^1H$ **NMR** (400 MHz, $CDCl_3$): $\delta$ 0.87 (t, DOPA $CH_3$, $J_{HH} = 7$ Hz, 18H), 1.25 (br, DOPA $CH_2$, 60H), 1.57 (br, DOPA $\alpha$ and $\beta$ $CH_2$, 24H); m/z (MALDI–ToF, matrix = trans-2-[3-(4-tert-butylphenyl)-2-methyl-2-properylidene] malononitrile, solvent = toluene): 1724.2 $\{[Zn_4O(DPPA)_5]^+$ calcd 1723.9} amu;

**Syntheses and characterization of 3A.** $Zn_6(\mu_2$-OH$)_3(DPPA)_9$: $ZnEt_2$ (100 mg, 0.809 mmol) was added to a toluene (8 ml) suspension of DPPA-H (265 mg, 1.21 mmol). After stirring for 30 min, a slight excess of water (10 µl, 0.56 mmol) in acetone (0.1 ml) was added. The resulting solution yielded a white precipitate, which could be isolated by removal of the solvent by evacuation (311 mg, 96% yield), which was deemed pure by powder XRD. Alternatively, single crystals of **3A** could be grown by gently warming the original toluene solution/suspension and then allowing it to cool to room temperature. $^{31}P\{^1H\}$ **NMR** (162 MHz, $CDCl_3$): $\delta$ 24.2 (s, 6P), 30.1 (s, 3P) p.p.m.; (162 MHz, $d_8$-THF): $\delta$ 23.7 (s, 6P), 29.2 (s, 3P) p.p.m.; $^1H$ **NMR** (400 MHz, 273 K, $CDCl_3$): $\delta$ 3.68 (s, OH, 3H), 6.57 (m, $C_6H_5$ 'planar (1)', 12H), 6.89 (t, $C_6H_5$ 'planar (1)', $J_{HH} = 7$ Hz, 6H), 6.94 (m, $C_6H_5$ 'planar (2)', 12H), 7.20 (t, $C_6H_5$ 'planar (1)', $J_{HH} = 7$ Hz, 6H), 7.3–7.52 (m, $C_6H_5$ 'bridging, planar (1) and (2)', 42H), 7.81 (dd, $C_6H_5$ 'bridging', $J_{HH} = 7$ and 12 Hz, 12H) p.p.m.; anal. calcd for (from toluene) $Zn_6P_9O_{21}C_{122}H_{109}.2(C_7H_8) = C$, 56.75; H, 4.25%; found C, 56.71; H, 4.36%.

**Syntheses and characterization of 3B.** $Zn_6(\mu_2$-OH$)_3(D^{MeO}PPA)_9$: **3B** formed as the minor product of an equilibrium with **2B**, when excess moisture is present in a hydrophobic solvent such as $CDCl_3$. For example, water (0.5 µl) was added to **2B** (9 mg, 0.0046 mmol) in $CDCl_3$ (0.5 ml). $^{31}P\{^1H\}$ **NMR** (162 MHz, $CDCl_3$): $\delta$ 24.8 (s, 6P), 30.6 (s, 3P) p.p.m.; $^1H$ **NMR** (400 MHz, $CDCl_3$): $\delta$ 3.61 (br, OMe, 18H), 3.70 (br, OMe, 18H), 3.77 (s, OMe, 18H), 3.81 (s, OMe, 3H), 6.22 (br $C_6H_4$OMe, 12H), 6.46 (br, $C_6H_4$OMe, 12H), 6.71 (m, $C_6H_4$OMe, 12H), 7.33 (m, $C_6H_4$OMe, $2 \times 12$H), 7.64 (m, $C_6H_4$OMe, 12H).

**Syntheses and characterization of 3C.** $Zn_6(\mu_2$-OH$)_3(DOPA)_9$: **3C** forms as the minor product of an equilibrium with **2C** when excess moisture is present. $^{31}P\{^1H\}$ **NMR** (162 MHz, $CDCl_3$): $\delta$ 49.9 (s, 6P), 53.9 (s, 3P); $^1H$ **NMR** (400 MHz, $CDCl_3$): $\delta$ 3.48 (s (OH), 3H), DOPA peaks overlap with **2C**.

**Syntheses and characterization of 4A.** $Zn_6B_3O_6(DPPA)_9$: DPPA-H (265 mg, 1.21 mmol) and B(OH)$_3$ (25 mg, 0.40 mmol) were placed in a Young's flask with a stirrer bar. To this, THF (10 ml) was added and $ZnEt_2$ (100 mg, 0.81 mmol) was then added dropwise, while stirring. The solution was stirred overnight. A small amount of white precipitate may form, which is expected to be **3A** from reaction with liberated water, the solution was thus filtered and precipitated using hexane to yield a white powder, which was dried under vacuum (200 mg, 60% yield). The bulk powder from rapid precipitation appears amorphous by powder XRD but single crystals could be grown by slow diffusion of hexane into a THF solution

of **4a**. $^{31}P\{^1H\}$ **NMR** (162 MHz, CDCl$_3$): $\delta$ 22.7 (s, 6P), 29.3 (s, 3P) p.p.m.; (162 MHz, d$_8$-THF): $\delta$ 22.5 (s, 6P), 29.0 (s, 3P) p.p.m.; $^1H$ **NMR** (400 MHz, CDCl$_3$): $\delta$ 6.79 (br, 'Asym' DPPA, 12H), 6.86 (br, 'Asym' DPPA, 6H), 7.21 (br, 'Asym' DPPA, 12H), 7.38 (m, 'Asym and Sym' DPPA, 36H), 7.55 (m, 'Asym' DPPA, 12H), 8.13 (m, 'Sym' DPPA, 12H); m/z (MALDI–ToF, matrix = trans-2-[3-(4-tert-butylphenyl)-2-methyl-2-properylidene] malononitrile, solvent = toluene): 2257.2, {[Zn$_6$O$_6$B$_3$(DPPA)$_8$]$^+$ calcd 2256.9} amu; anal. calcd for: Zn$_6$P$_9$B$_3$O$_{18}$C$_{114}$H$_{102}$ = C, 55.58; H, 4.17%; found C, 55.32; H, 3.94%.

**Syntheses and characterization of 5A.** Zn$_{11}$Et$_{10}$O$_4$(DPPA)$_4$: DPPA-H (96.4 mg, 0.442 mmol) was placed in a Schlenk flask, with a stirrer bar. To this, toluene (10 ml) was added and ZnEt$_2$ (150 mg, 1.21 mmol) was then added dropwise, while stirring. To this solution, water (7.95 μl, 0.442 mmol) was added, by Eppendorf syringe, under a stream of N$_2$. The solution was stirred overnight, to ensure all water had been dissolved. Hexane was added and the product isolated as crystals (isolated yield = 77 mg, 36%). $^{31}P\{^1H\}$ **NMR** (162 MHz, CDCl$_3$): $\delta$ 33.1 (s, 4P) p.p.m.; (162 MHz, C$_6$D$_6$): $\delta$ 33.7 (s, 4P) p.p.m.; $^1H$ **NMR** (400 MHz, CDCl$_3$): $\delta$ −1.48 (q, CH$_2$, $J_{HH}$ = 8 Hz, 4H), − 0.07 (dm, CH$_2$*, 16H), 0.23 (q, CH$_3$, $J_{HH}$ = 8 Hz, 6H), 1.24 (t, CH$_3$, $J_{HH}$ = 8 Hz, 24H), 7.41 (m, DPPA, 8H), 7.50 (m, DPPA, 16H), 7.98 (m, DPPA, 16H) p.p.m.; anal. calcd for: Zn$_{11}$P$_4$O$_{12}$C$_{68}$H$_{90}$ = C, 42.05; H, 4.67%; found C, 42.11; H, 4.52%. *Symmetrical multiplet, suggesting diastereotopic Et CH$_2$ protons. Both halves of multiplet couple to the CH$_3$ signal ($\delta$ 1.24) in the $^1H$-$^1H$ correlation spectroscopy spectrum.

**Syntheses and characterization of 5C.** Zn$_{11}$Et$_{10}$O$_4$(DOPA)$_4$: (64.1 mg, 0.221 mmol) of dioctylphosphinic acid was placed in a Young's tap flask with a stirrer bar. To this, 10 ml of toluene was added and 75 mg (0.607 mmol) of ZnEt$_2$ was then added dropwise, while stirring. To this solution, 3.9 μl (0.217 mmol) of water was added by Eppendorf syringe under a stream of N$_2$. The solution was stirred for 1 h followed by brief sonication to ensure all water had been incorporated. The product was isolated by evacuation of the solvent to give an air-sensitive colourless oil. $^{31}P\{^1H\}$ **NMR** (162 MHz, CDCl$_3$): $\delta$ 60.8 (s, 4P); (162 MHz, d$_8$-toluene): $\delta$ 61.1 (s, 4P) (162 MHz, h$_8$-toluene): $\delta$ 61.1 (s, 4P); $^1H$ **NMR** (400 MHz, CDCl$_3$): $\delta$ 0.14 (q, Et CH$_2$, $J_{HH}$ = 8 Hz, 16H), 0.45 (q, Et CH$_2$, $J_{HH}$ = 8 Hz, 4H), 0.89 (br, DOPA CH$_3$, 24H), 1.14 (t, Et CH$_3$, $J_{HH}$ = 8 Hz, 24H), 1.2–1.9 (m, DOPA CH$_2$, 112H); (400 MHz, h$_8$-toluene): $\delta$ 0.72 (q, Et CH$_2$, $J_{HH}$ = 8 Hz, 16H), 0.91 (t, DOPA CH$_3$, 24H), 0.95 (q, Et CH$_2$, $J_{HH}$ = 8 Hz, 4H), 1.29–1.52 (br, DOPA CH$_2$, 112H), 1.67 (t, Et CH$_3$, $J_{HH}$ = 8 Hz, 24H), *CH$_3$ signal for the minor ethyl group is not located, probably obscured.

**Data availability.** The data supporting the findings of this study are available within the article and its Supplementary Information or are available from the authors. The Crystallographic data have been deposited with the Cambridge Crystallographic Data Centre under CCDC 1432882–1432886. These data can be obtained free of charge from The Cambridge Crystallographic Data Centre via www.ccdc.cam.ac.uk/data_request/cif. Full bond length and bond angle data may be found in the CIFs, which are available as Supplementary Data 1–5.

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

## Acknowledgements
The EPSRC are acknowledged for funding (EP/K035274/1, EP/M013839/1 and EP/H046380/1).

## Author contributions
S.D.P. conducted all experimental work. E.R.W. conducted all TEM characterization and associated analysis. S.D.P., M.S.P.S and C.K.W. conceived the experiments. All authors were involved in writing the manuscript.

## Additional information

**Competing financial interests:** The authors declare no competing financial interests.

