## [Peer Review File · Nature Communications]

Reviewers' comments:

Reviewer #1 (Remarks to the Author):

What are the major claims of the paper?

The authors report the reactions of Et_2Zn with phosphinic acids in the presence of various amounts of water. The controlled hydrolysis steps result in the formation of ZnO cage-like compounds of varying size and nuclearity; the increase in size is believed to provide insight into the way in which ligand-capped ZnO nanoparticles form. The molecular clusters are thought to form cleanly, and they have been characterized in the solid-state by single-crystal XRD, and by ^1H and ^{31}P NMR spectroscopy.

Are they novel and will they be of interest to others in the field?

The most novel development of the paper is that step-wise controlled hydrolysis of ZnEt_2 can be achieved with phosphinic acid pro-ligands, resulting in the formation of isolable molecular cage compounds. A huge amount has already been done with carboxylates, so it's a question of whether or not phosphinate ligands represent a sufficiently new direction to merit publication in NComms.

Is the work convincing, and if not, what further evidence would be required to strengthen the conclusions?

One concern I have with this study is that the yields reported for the new compounds are low, i.e. 20-40%. What happened to the remaining material? Although the isolated materials seem pure, it is possible that they have been crystallized as the least soluble component of a more complicated mixture. I note also that the materials 1-5 have only been characterized in the solid state by single-crystal XRD. Some bulk characterization of the solid would have been useful, e.g. powder XRD. One can determine the powder pattern of materials 1-5 from the single-crystal measurements, hence a comparison of that with the powder pattern of a bulk sample would give insight into the presence of additional compounds. CHN analysis doesn't help much because it only confirms the relative amounts of elements, i.e. dimers are not distinguished from tetramers, etc. The bulk material has been characterized in solution by ^{31}P NMR spectroscopy; why not solid-state NMR?

Following on from the above comments, the synthesis of the Zn11 species produced a ^{31}P spectrum (of the reaction mixture) with multiple low-intensity peaks. What could they be due to?

The bond lengths and angles are not stated with esd values. Why? Even though the structure of 2A is not great, we still need some data. Provide the cif file.

I was surprised that the water content of the 'dry' solvents was not first determined using a K. Fischer kit. Given that small amounts of water can seemingly tip the balance away from one structure towards another, this information would have been useful.

Most of the Figures are too small, particularly 2 and 3.

On a more subjective note, do you feel that the paper will influence thinking in the field?

Hard to say. There is undoubtedly a lot of interest in ZnO nanoparticles and their properties, but this paper does look a bit like crystal fishing. I need more convincing evidence for the formation of materials 1-5.

Overall, I recommend:

Do not accept the paper yet. Substantial revisions are required.

Improve the yields or account for the fact that most of the material doesn't crystallize.

Powder diffraction on bulk samples.

Solid-state NMR on bulk samples.

The manuscript is also verbose in parts. For example, in the abstract, why not write "31P NMR spectroscopy" instead of "the 31P NMR spectroscopic handle"?

Reviewer #2 (Remarks to the Author):

In their manuscript entitled „Simple Phosphinate Ligands Access New Zinc Clusters Identified in the Synthesis of ZnO Nanoparticles", Milo S. P. Shaffer, Charlotte K. Williams and co-workers report on a series of polynuclear zinc oxide-based complexes or clusters, which were accessed by reactions of ZnEt₂ with diverse ligand compounds in stoichiometric or substoichiometric amounts. At first glance, the work is located in the very popular field of ZnO nanoparticle synthesis by hydrolysis of organo-zinc compounds at room temperature in organic solvents, which was introduced by the Chaudret group in 2005, and to which Shaffer and Williams and co-workers have also contributed recently. Here, they focus on the formation pathway from the molecular precursors to the nanoparticles. Upon further reading one recognizes, however, that the report is mainly about synthesis and characterization of zinc oxide complexes and clusters that can be prepared alongside of nanoparticles and seem to be independent from the latter (see the more detailed comments below).

The investigations were set out with the ligand diphenylphosphinic acid (DPPA-H; compound series A), which the group has previously shown to be suitable for nanoparticle synthesis. In addition, they utilized bis(4-methoxyphenyl)phosphonic acid (DMeOPPA-H; compound series B) and dioctylphosphonic acid (DOPA-H; compound series C), which is a very commonly used shielding ligand in nanoparticle synthesis.

The different compounds were obtained upon reaction of the precursor ZnEt₂ with different amounts of the ligand, in the presence or absence of water or B(OH)₃, or by further treatment of (some of the) compounds with water or B(OH)₃. The authors were able to detect five different types of complexes or clusters (consecutively numbered 1 - 5), which were found as a complete series for ligand type (series) A, while for series B and C, one of the compounds is not presented that forms (only) in the presence of B(OH)₃. Compounds 1A, (2A), 3A, 4A, 5A and 2B were also obtained in single-crystalline form (with the data of 2A currently not allowing for a full structure refinement). Compounds 1B, 5B, 1C and 5C were only identified by means of NMR spectroscopy (¹H and ³¹P), all other compounds were detected and characterized by NMR spectroscopy and MALDI mass spectrometry.

NMR studies have proven very powerful here as 1 - 5 produce only one (1, 2, 5) or two (3, 4) characteristic ³¹P NMR signals. Hence, NMR studies were also undertaken to get an insight in the product mixtures that evolve from different reactant stoichiometries (especially regarding the amount of water added) and to further investigate the equilibrium observed between compounds 2 and 3. Finally, the authors monitored the formation of ZnO nanoparticles by in situ NMR and UV-spectroscopy studies.

As a general statement on the work, I would like to express my appreciation for the authors' attempts to get and provide insight into the complex processes of nanoparticle growth. To the best of my knowledge, most of the findings, in particular in the way they were put together, are new. They may help to understand the very first steps of the formation of larger (L)ZnO(H) aggregates prior to, alongside, or even independent from nanoparticle formation, although I do not agree with the suggested role of the compounds in the light of nanoparticle synthesis. As correctly stated by the authors during their discussion, the L₆Zn₄O motif of compound 2 is very commonly found for carboxylate ligands (very prominent in MOFs), but it was not yet observed with phosphinate ligands. I agree with most of the general conclusions on the experimental finding, including the equilibrium between compounds 2 and 3.

However, on the other hand, several drawbacks of the report need to be mentioned and considered when judging on its value. The major points of criticism regarding the contents are the following:

a) First and foremost, I am not convinced so far that the compounds really represent precursors,

hence "seeds", for nanoparticles, which is quite obvious if one considers the low structural match between all of the structures (except that of 2), especially of the predominant species 5, and any Wurtzite-type ZnO nanoparticle. The structure of 2 is clearly related to the ZnO lattice of nanoparticles, but this species is correctly considered as being a (competing?) by-product, as it occurs at the same time as the nanoparticles themselves. All other compounds will rather form independently, and even in competition to the nanoparticles. Some of the reactions are naturally independent for nanoparticle synthesis, such as the reaction in the presence of B(OH)₃, which is, although interesting, an independent branch of the study. Thus the context of the work is not well-chosen in my view, although the last part deals with (well-established and therefore not very original) nanoparticle synthesis (see also below).

b) In agreement with the former point, the main part of the report is presented as a "versatile and accessible route to new zinc clusters", which is clearly the more appropriate way to present it, but at the same time, this reduces its overall and broad meaningfulness, of course. Still, I prefer a well-measured presentation of the results as they stand, without the need of using any buzz words or teasing expressions, such as "spontaneous self-assembly processes which rapidly adjust to external stimuli" (to be translated as: "one does not know so far how the complexes form, but we can obviously influence it by addition of reactants or changing external conditions, as usual") or "some very unusual, yet generally applicable larger structures are identified" (it remains unclear what the authors wish to use the compounds for; mind that one won't use "structures", but rather "compounds" anyway). Also the conclusion section ought to be revised in this sense.

c) The authors talk about "equilibria" or "interconversion" of the species in solution, which I would rather call co-existence in most cases. A real interconversion, i.e., a reaction backwards and forwards, was only (though nicely!) proven for the equilibrium between 2 and 3, whereas in all other cases, different reactant stoichiometries led to different relative amounts of products, which can also happen if their formation pathways are completely independent from each other. I cannot see a reaction herein (besides the mentioned pair of compounds 2 and 3, and the independent study on the formation of 4), where a compound was clearly formed out of another one. If so, this should be said more clearly. If not, some of the equilibrium arrows have to be removed from Figures 1 and 5, and the discussion and conclusion paragraphs need to be adjusted accordingly.

d) Partial incompleteness of the results regarding the existence and/or mode of identification of the diverse species with different ligands, as well as (in part) different synthetic accesses to some of the compounds, complicates the reading and comprehension of the full study. One might concentrate the report on one series of compounds, or try and complete the other series as well.

Minor points of criticism are the following:

a) The crystallographic data of compounds 3A, 4A and 5A suffer from a large number of restraints, which for 5A is dramatic as it exceeds the number of parameters by a factor of >2 here. This should be given another round of X-ray diffraction analyses.

b) The rather elaborate report on the nanoparticle synthesis (page 6 of my copy) is dispensable in my view; it is in most parts a repeat of known literature, the corresponding TEM figures are not at all delighting, and the correlation with the series of new compounds remains questionable, as outlined above. One might summarize the findings illustrated in Figure 6 in a short paragraph as an addition to the report on the clusters.

c) The manuscript contains some inappropriate expressions, such as "pure metal cluster complexes" or "bimetallic clusters" where ligand-bridged complexes or clusters are meant, and some typos (c.f. "μ" not printed on page 5). The second paragraph of the introduction seems to be somewhat unorganized regarding the order of mentioned types of (existing/non-existing) compounds.

In conclusion, I recommend thorough revision of the work according to the issues raised above. Further, I am not convinced that Nature Communications will be the appropriate platform for communicating the findings, which are definitely nice but presumably not in the position to attract the broad readership of Nature Communications upon focusing on the relevant results. I see it more appropriately published in a chemistry or materials science journal.

Reviewer #3 (Remarks to the Author):

The present article is a very thorough analysis of the chemical processes related to ZnO formation on protolysis/hydrolysis of zinc alkyl compounds. The use of diorganophosphinic acids as stabilizing ligands plays a key role in this study, since various ZnO clusters and also nanoparticles can be supported by this ligand, and the ^{31}P NMR signal of the PO₂ group seems to be sensitive enough to its chemical environment allowing a versatile in-situ distinction of different species by ^{31}P NMR spectroscopy. In general the work has been performed with great care and the scientific argumentation is sound throughout the whole article. However, some important questions remain unanswered and should be addressed more clearly in my opinion:

1. The in-situ hydrolysis experiment of ZnO NP formation monitored by ^1H and ^{31}P NMR spectroscopy is indeed very interesting and most central to the article. In my understanding, the formation of 2 at high water levels, which occurs clearly parallel to the NP formation, is well explained by the fact that 2 contains the highest DOPA:Zn ratio of all identified clusters, i.e. it serves as a "ligand reservoir" of excess DOPA, which is liberated on agglomeration of ZnO units. However, by increasing the water level above 100%, I would expect formation of 3, which is not mentioned in the text. Also the formation of free DOPA-H cannot be excluded, which might show a fast ligand exchange with coordinated DOPA preventing its observation in in-situ NMR experiments. Additional experiments, e.g. VT-NMR of DOPA stabilized ZnO NP in presence of excess DOPA-H or the use of deuterated DOPA-D/D₂O are recommended.

2. The reaction scheme shown in Figure 6a is in my opinion misleading, since the stoichiometry presented in the equations is not fully supported by the experiments. I recommend to include the area graph S48 (%P vs %hydrolysis) instead, which is really helpful in understanding the complex interplay of the various chemical processes involved.

3. ZnO NPs are most often prepared by a sol/gel approach with different types of stabilizing agents, carboxylic acids or carboxylates being amongst the most important ones. The similarities/differences of carboxylates and phosphinates need to be addressed and discussed with more care. E.g. a comparison of the chemical stability of 2 and its well known carboxylate counterpart, especially their hydrolytic stability, might help to convince the reader of the scientific relevance of this study. The sentence "[phosphinates] are more acidic and so stronger binding to zinc is expected" is counterintuitive (phosphinic acids are more acidic than carboxylic acids) and should be discussed/explained.

4. While the characterization of the molecular species presented in the article is excellent, the characterization of the ZnO NP is much less conclusive. E.g. TEM images are only given for the "yellow ZnO" but not for the final NP after complete hydrolysis. Why? The results from absorption UV spectroscopy are not convincing, recording of fluorescence spectra is recommended. In general, a more thorough analysis of the final hydrolysis product, i.e. the DOPA capped ZnO NPs, is needed.

5. My major concern is the presentation of the article. While the molecular clusters including the hydrolysis experiment are very thoroughly discussed, the article is less convincing when NP chemistry is concerned. However, the scientific relevance of this study is in my opinion strongly based on its significance for ZnO NP synthesis: Where exactly is the scientific impact of an elucidation of the mechanistic details of ZnO NP formation? I do not doubt the importance of this study, but a thorough discussion of the state of the art of ZnO NP synthesis, its open questions and problems and discussion of the significance of the present study in the light of these problems and questions would in my opinion strongly enhance the scientific impact of this paper.

In general, I am convinced that the study is highly interesting and the results most relevant to a large chemical community. I recommend acceptance of this article in "Nature Communications" after thorough revision and consideration of the points raised above.

RESPONSE TO THE REVIEWERS COMMENTS FOR:

Simple Phosphinate Ligands Access New Zinc Clusters Identified in the Synthesis of ZnO Nanoparticles.

Sebastian D. Pike, Edward R. White, Milo S. P. Shaffer,* Charlotte K. Williams*

We would like to thank all three reviewers for their thorough guidance and feedback: we appreciate the time taken and the insights provided. We have conducted additional experiments and have made changes to the manuscript and supporting information: the comments are now addressed in full and we feel that the manuscript is substantially improved. Listed below are point-by-point responses to the comments and questions.

Reviewer #1 (Remarks to the Author):

What are the major claims of the paper?

The authors report the reactions of Et_2Zn with phosphinic acids in the presence of various amounts of water. The controlled hydrolysis steps result in the formation of ZnO cage-like compounds of varying size and nuclearity; the increase in size is believed to provide insight into the way in which ligand-capped ZnO nanoparticles form. The molecular clusters are thought to form cleanly, and they have been characterized in the solid-state by single-crystal XRD, and by ^1H and ^{31}P NMR spectroscopy.

Are they novel and will they be of interest to others in the field?

The most novel development of the paper is that step-wise controlled hydrolysis of ZnEt_2 can be achieved with phosphinic acid pro-ligands, resulting in the formation of isolable molecular cage compounds. A huge amount has already been done with carboxylates, so it's a question of whether or not phosphinate ligands represent a sufficiently new direction to merit publication in NComms.

Whilst the review correctly notes that there is a significant body of literature on the study of zinc carboxylate clusters as well as quite a range of different applications for them, the current study represents far more than an incremental advance in knowledge using phosphinates. In fact, the phosphinate ligands, although isoelectronic with carboxylates, provide unique insights through in situ monitoring of speciation, interconversion and nanoparticle (NP) formation: such insight is simply not possible using carboxylate ligands. The use of phosphinate ligands allows solution based ^{31}P NMR spectroscopy to be used to characterize product mixtures, confirm formation of pure individual clusters with defined nuclearities, interrogate equilibria between clusters (including in combination with ^1H NMR to determine thermodynamic parameters) and study the formation of ligated zinc oxide nanoparticles. It is important to note that such equilibria and mixtures of clusters may well also be relevant to zinc carboxylate clusters but cannot be easily interrogated by only ^1H NMR spectra due to complexity and lower sensitivity to cluster conformation. The new and previously inaccessible understanding gained using zinc-phosphinate clusters, is thus of potential general relevance to the preparation of ligated nanoparticles, and sheds light on the missing links between molecular precursors and nanoparticles. This area is poorly understood in the literature, due to a lack of tools to probe the relevant species. Further, the identification of stable products, during the reaction trajectory, has allowed us to then directly synthesise these products in pure form. The work in our paper, using phosphinate ligands, highlights a general approach that can be applied to such systems. As an illustration of the power of this approach, our study has revealed a highly unexpected and counterintuitive result, namely that the nanoparticles initially grow as bare ZnO, and are only ligated at the

end of the synthesis. Understanding this process in more detail offers rich opportunities both to modify properties and to develop new theory. We wish to emphasise that there is not yet any unifying understanding of how simple carboxylate clusters interconvert, despite their undoubted importance and prevalence. In short, this study provides both a new approach and new understanding of important set of reactions; it is not reasonable to consider it as a simple 'additional' example of a ligand related to carboxylate.

Is the work convincing, and if not, what further evidence would be required to strengthen the conclusions?

One concern I have with this study is that the yields reported for the new compounds are low, i.e. 20-40% What happened to the remaining material? Although the isolated materials seem pure, it is possible that they have been crystallized as the least soluble component of a more complicated mixture.

The original reported yields were for isolated crystals and so values of 20-40% are as expected for small-scale reactions conducted under anaerobic conditions. Nonetheless, we provide unambiguous NMR evidence, all of which was already provided in the ESI, which clearly demonstrates that the clusters are either formed as the only product (for **1A**, **2A**, **2B**, **4A**, **5A**, **2C**) or as part of equilibria with one another (in the case of: **2A/3A**; **2B/3B**; **2A/3A/4A**; **5A/2A/1A**; **2C/3C**; **5C/2C/1C**). We wish to emphasise that this NMR data is truly representative of all the phosphinate-containing species present as the measurements are conducted in situ, without any purification or modification on the homogeneous reaction solutions. Thus, we can be confident that the quantitative conversion to the characterized complexes occurs. In order to reassure the reviewer that significantly greater isolated yields can be obtained, in line with the quantitative conversion identified by NMR, we have conducted additional experiments on larger scale so as to reduce losses on sample isolation. The isolated yields are now:

2A = 83% yield; **2B** = 80% yield; **3A** = 96% yield; **4A** = 60% yield.

These high yields were obtained having identified the molecular stoichiometry using NMR spectroscopy during initial experiments, highlighting the utility of in-situ solution studies to allow rational design of synthetic procedures to give pure products.

These new preparation routes, along with the improved yields are included in the supporting information p.S2-S6

I note also that the materials 1-5 have only been characterized in the solid state by single-crystal XRD. Some bulk characterization of the solid would have been useful, e.g. powder XRD. One can determine the powder pattern of materials 1-5 from the single-crystal measurements, hence a comparison of that with the powder pattern of a bulk sample would give insight into the presence of additional compounds. CHN analysis doesn't help much because it only confirms the relative amounts of elements, i.e. dimers are not distinguished from tetramers, etc. The bulk material has been characterized in solution by ³¹P NMR spectroscopy; why not solid-state NMR?

Whilst, as noted above, the NMR indicates that no other cluster species are present in solution, we agree that powder XRD patterns can provide a useful confirmation that no other structures occur in the solid state. For this reason, we have applied powder XRD to the complete crude products of the reactions, for samples **2a** (now added to the supporting information as Fig S9) and **3a** (now added as Fig. S19). The powder XRD measurements show excellent agreement with simulated patterns from the single crystal data, thereby confirming the NMR measurements and the bulk purity in the solid-state. Despite showing

quantitative conversion by solution NMR spectroscopy, the powder sample of **4A** produced only an amorphous XRD pattern (Fig S31), with broad signals in the expected areas. As is common for larger, more complex molecules, it may be that fast precipitation routes hinder crystallisation (note that the isolated crystals were obtained by slow crystallisation methods).

We do not believe that solid-state NMR, in particular ^{31}P solid-state NMR, would be likely to provide any new information. The bulk solid-state phases purity are already established by elemental analysis and now by XRD; solution NMR spectroscopies confirm these data and can usefully be applied in situ during the reactions. To use solid-state NMR misses the point and benefit of our strategy, which focuses on homogeneous solution synthesis of clusters and nanoparticles. We also note that ^{31}P magic angle spinning solid-state NMR can result in broadened signals - especially when multiple crystallographically independent ^{31}P centres are present and where low degrees of mobility within the solid state are expected (such as in our crystalline materials) (for related crystalline ^{31}P SSNMR examples see *J. Am. Chem. Soc.*, **2015**, 137 (2), pp 820–833); it is therefore unlikely that such experiments would improve on the data already collected, or contribute to additional understanding.

Following on from the above comments, the synthesis of the Zn11 species produced a ^{31}P spectrum (of the reaction mixture) with multiple low-intensity peaks. What could they be due to?

The reviewer is referring to the NMR spectrum shown in Fig S54 which does contain some low-intensity, unidentified signals as is clearly discussed in the supporting information. In this case, the air-sensitive compound **5C** proved particularly difficult to isolate and we believe that minor imbalances in the reagent stoichiometry together with the presence of trace O_2 has resulted in the formation of these minor species. Whilst it has not proved possible to identify all the minor species observed in the isolated NMR spectrum of **5C**, it is worth noting that **5A** can be isolated without any contaminants (Fig S33, S34). It may be that using the long-chain DOPA-H leads to more difficulties in removing solvent residues etc. It is also important to note that the quantitative and selective formation of **5C** is achieved during the in-situ monitoring of the synthesis of $\text{ZnO}@$ DOPA NPs (now shown in Fig S49, in particular at 75% hydrolysis).

The bond lengths and angles are not stated with esd values. Why?

We apologise for this oversight and have added errors to all our data which is now included to 3 decimal places in the text.

Even though the structure of **2A** is not great, we still need some data. Provide the cif file.

The structure of **2A** (collected twice from different solvent systems) is relatively poor likely due to its spherical shape which may hinder crystallisation in a well-defined conformation (for a related example see *Organometallics*, 2015, 34, p 1487). The structure is included in the supporting information to verify the connectivity of the heavy atoms (showing consistency with structure **2B**) and now to confirm phase purity in comparison with the very clean powder XRD data. We have added some clarifying points to the ESI regarding the structure of **2A** along with unit cell parameters, we do not believe it is appropriate to deposit the CIF in the relevant databases, however, we are including the CIF of the incomplete structure, with this response, in case it is helpful for the further review process.

I was surprised that the water content of the 'dry' solvents was not first determined using a K. Fischer kit. Given that small amounts of water can seemingly tip the balance away from one structure towards another, this information would have been useful.

Water content has been tested for the solvents used and the relevant data has been added to the supporting information. The solvents contain moisture levels <5 ppm in all cases.

Most of the Figures are too small, particularly 2 and 3.

Figs 1, 2, 3 and 4 have been redesigned.

On a more subjective note, do you feel that the paper will influence thinking in the field? Hard to say. There is undoubtedly a lot of interest in ZnO nanoparticles and their properties, but this paper does look a bit like crystal fishing. I need more convincing evidence for the formation of materials 1-5.

We do not feel that 'crystal fishing' is a fair criticism of this work. Indeed, all the new complexes are fully characterized in solution by a range of NMR spectroscopies as well as by bulk solid state and crystallographic methods. Furthermore, it is exactly because we have been able to monitor the evolution of the homogeneous reaction system that we were able to design direct, clean and quantitative syntheses of the particular complexes identified. The crude NMR data are particularly convincing with regard to both complex speciation and purity. Indeed, the whole point of using NMR-active ligands, such as these phosphinates, is to demonstrate that the field can move well beyond 'crystal fishing' to a more detailed understanding of the structures observed under particular conditions and their interconversion. We are keen to avoid any such misinterpretation by both reviewers and readers and so have moved Fig. S48 to Fig. 6c in the revised manuscript. This figure clearly demonstrates the composition of the entire system as hydrolysis progresses and shows the evolution of the species from well-defined diethyl zinc + phosphinic acid precursors, through characterized cluster compounds, to the ligated ZnO nanoparticles.

Overall, I recommend:

Do not accept the paper yet. Substantial revisions are required.

Improve the yields or account for the fact that most of the material doesn't crystallize.

Powder diffraction on bulk samples.

Solid-state NMR on bulk samples.

We have substantially modified the manuscript and refer the reviewer to the detailed solution NMR studies and the newly collected bulk XRD data from batches obtained in high yields.

The manuscript is also verbose in parts. For example, in the abstract, why not write "31P NMR spectroscopy" instead of "the 31P NMR spectroscopic handle"?

We have tried to make the manuscript as concise as possible. We have changed the specific phrase in places in the manuscript but this wording was chosen exactly to emphasise the conceptual importance of using phosphinates. Their role as a unique, head group based, NMR handle is the key enabler of the study, both for in situ monitoring, and rational design of direct syntheses. It is important to emphasise, for readers new to such ligands, the distinct benefit and difference to common alternative ligands (carboxylates or amines).

Reviewer #2 (Remarks to the Author):

In their manuscript entitled „Simple Phosphinate Ligands Access New Zinc Clusters Identified in the Synthesis of ZnO Nanoparticles“, Milo S. P. Shaffer, Charlotte K. Williams and co-workers report on a series of polynuclear zinc oxide-based complexes or clusters, which were accessed by reactions of ZnEt₂ with diverse ligand compounds in stoichiometric or substoichiometric amounts. At first glance, the work is located in the very popular field of ZnO nanoparticle synthesis by hydrolysis of organo-zinc compounds at room temperature in organic solvents, which was introduced by the Chaudret group in 2005, and to which Shaffer and Williams and co-workers have also contributed recently. Here, they focus on the formation pathway from the molecular precursors to the nanoparticles. Upon further reading one recognizes, however, that the report is mainly about synthesis and characterization of zinc oxide complexes and clusters that can be prepared alongside of nanoparticles and seem to be independent from the latter (see the more detailed comments below).

The term 'independent' is indeed a good observation, as the data demonstrate that the interconnected cluster complexes form separately to the growing ZnO NPs (as stated in the text). The very surprising but clear result of the in situ monitoring, is that the bare nanoparticles sequester the clusters to form ligand-terminated ZnO only in the final step. This mechanism is quite different to the usual assumptions. To demonstrate this unexpected finding, it was essential to fully characterise the cluster equilibria before considering the nanoparticle synthesis.

The investigations were set out with the ligand diphenylphosphinic acid (DPPA-H; compound series A), which the group has previously shown to be suitable for nanoparticle synthesis.

To clarify, DPPA-H has not been previously shown to be suitable for NP synthesis. The ligand DOPA-H is the previously studied ligand, as stated in the text. The DPPA-H series is required to establish the cluster structures, as the phenyl ligand facilitates crystallisation, confirming the NMR interpretation used to analysis the DOPA-H phenomenology.

In addition, they utilized bis(4-methoxyphenyl)phosphonic acid (DMeOPPA-H; compound series B) and dioctylphosphonic acid (DOPA-H; compound series C), which is a very commonly used shielding ligand in nanoparticle synthesis.

To clarify, the pro-ligands used were all phosphinic acids and not phosphonic acids. The structure is quite different, with two alkyl chains and a different head group. Definitions of phosphinates, phosphonates and dialkylphosphates are included in the re-worded introduction. We cannot accept the statement that DOPA-H is a 'very commonly used shielding ligand': a 'Scifinder' search of 'DOPA-H and DOPA⁻' reveals only 12 papers and 4 patents concerning any nanomaterial combined with DOPA, mainly concerning CdSe. In contrast, trioctylphosphine oxide (TOPO) and various phosphonic acids are widely applied. Nonetheless the processes and findings here may be applied to P containing ligands more generally.

The different compounds were obtained upon reaction of the precursor ZnEt₂ with different amounts of the ligand, in the presence or absence of water or B(OH)₃, or by further treatment of (some of the) compounds with water or B(OH)₃. The authors were able to detect five different types of complexes or clusters (consecutively numbered 1 - 5), which were found as a complete series for ligand type (series) A, while for series B and C, one of the compounds is not presented that forms (only) in the presence of B(OH)₃.

We refer to the more detailed discussion below and note that compound 4 is not relevant to ZnO nanoparticle formation. It does serve to demonstrate the broader utility of such clusters,

and to illustrate the equilibria present, but given its lower relevance to nanoparticles it was only studied using DPPA as the ligand (4A).

Compounds 1A, (2A), 3A, 4A, 5A and 2B were also obtained in single-crystalline form (with the data of 2A currently not allowing for a full structure refinement). Compounds 1B, 5B, 1C and 5C were only identified by means of NMR spectroscopy (^1H and ^{31}P), all other compounds were detected and characterized by NMR spectroscopy and MALDI mass spectrometry.

NMR studies have proven very powerful here as 1 - 5 produce only one (1, 2, 5) or two (3, 4) characteristic ^{31}P NMR signals. Hence, NMR studies were also undertaken to get an insight in the product mixtures that evolve from different reactant stoichiometries (especially regarding the amount of water added) and to further investigate the equilibrium observed between compounds 2 and 3. Finally, the authors monitored the formation of ZnO nanoparticles by in situ NMR and UV-spectroscopy studies.

As a general statement on the work, I would like to express my appreciation for the authors' attempts to get and provide insight into the complex processes of nanoparticle growth. To the best of my knowledge, most of the findings, in particular in the way they were put together, are new. They may help to understand the very first steps of the formation of larger (L)ZnO(H) aggregates prior to, alongside, or even independent from nanoparticle formation, although I do not agree with the suggested role of the compounds in the light of nanoparticle synthesis.

We thank referee two for their kind comments, but we are not sure with which suggested role the reviewer disagrees specifically? Nonetheless, we have discussed our findings further below which may help to clarify our interpretation.

As correctly stated by the authors during their discussion, the $\text{L}_6\text{Zn}_4\text{O}$ motif of compound 2 is very commonly found for carboxylate ligands (very prominent in MOFs), but it was not yet observed with phosphinate ligands. I agree with most of the general conclusions on the experimental finding, including the equilibrium between compounds 2 and 3.

However, on the other hand, several drawbacks of the report need to be mentioned and considered when judging on its value. The major points of criticism regarding the contents are the following:

a) First and foremost, I am not convinced so far that the compounds really represent precursors, hence "seeds", for nanoparticles, which is quite obvious if one considers the low structural match between all of the structures (except that of 2), especially of the predominant species 5, and any Wurtzite-type ZnO nanoparticle.

We did not describe the Zn cluster compounds as "seeds" to NPs, and agree that there is no obvious structural match to Wurtzite. On the contrary, our data shows that the clusters are important in the distribution of the ligands, particularly at the end of the synthesis; there is no evidence that the ligated clusters nucleate the nanoparticles, and we are not suggesting it. Overall, our findings are quite distinct to the usual assumed models. We have tried to clarify this point further in the manuscript throughout, and particularly in the final pages of discussion (p.7,8) where we now state explicitly "the cluster compounds do not obviously relate to Wurtzite and instead appear to act only as a reservoir of ligand "

The term "seeds" was used twice in the original text, but only in the introduction to describe research by other groups using metal clusters. However to avoid any confusion, the term "seeds" is removed from the introduction and replaced with 'building units' which was the descriptor applied in reference 28 to describe those compounds.

The structure of **2** is clearly related to the ZnO lattice of nanoparticles, but this species is correctly considered as being a (competing?) by-product, as it occurs at the same time as the nanoparticles themselves. All other compounds will rather form independently, and even in competition to the nanoparticles.

We completely agree with the reviewer's analysis of the data: it is surprising that the clusters play these roles but this phenomenology is clearly established by our in situ methodology. The "independent" formation of clusters at the beginning of the hydrolysis pathway sequesters all the available ligand before nanoparticle nucleation. During nanoparticle formation, the clusters and nascent NPs coexist but as separate entities until the hydrolysis is complete. As the hydrolysis is completed, cluster **5C** reacts leading to the formation of well-defined ZnO@DOPA NPs. We hope that this point is now clear in the revised manuscript, particularly with the inclusion of Fig. 6c. As referee three suggests, the formation of **2C** (which has a high ligand:Zn ratio) is consistent with the release of ligands from stable **5C** at the end of hydrolysis.

Some of the reactions are naturally independent for nanoparticle synthesis, such as the reaction in the presence of B(OH)₃, which is, although interesting, an independent branch of the study. Thus the context of the work is not well-chosen in my view, although the last part deals with (well-established and therefore not very original) nanoparticle synthesis (see also below).

We accept that the reaction with B(OH)₃ (to form **4A**) is separate to any discussion regarding nanoparticles. Nonetheless, **4A** is included because it demonstrates the ability to use the clusters to prepare mixed element complexes (which may even be relevant for doped nanoparticle work in future) and it provides further insight into the equilibrium interconversions between Zn(oxo) clusters: **4A** + H₂O are in equilibrium with **2A/3A**, see Fig S32. We already had some considerable discussion ourselves about whether to include this species in the manuscript, but decided on balance that the reactions were helpful. However, we would be happy to follow the editors' advice on this matter.

We agree that the basic synthesis of ZnO nanoparticles by the hydrolysis of ZnEt₂ under mild conditions has become established. Indeed, the method is being applied across a range of fields and looks to be particularly useful and complementary to more traditional nanoparticle syntheses. However, the key point is that there remains hardly any understanding of what happens at a molecular level during the hydrolysis reaction. This mechanistic insight is the primary novelty and importance of the study, including the surprising cluster behaviour noted above.

b) In agreement with the former point, the main part of the report is presented as a "versatile and accessible route to new zinc clusters", which is clearly the more appropriate way to present it, but at the same time, this reduces its overall and broad meaningfulness, of course. Still, I prefer a well-measured presentation of the results as they stand, without the need of using any buzz words or teasing expressions, such as "spontaneous self-assembly processes which rapidly adjust to external stimuli" (to be translated as: "one does not know so far how the complexes form, but we can obviously influence it by addition of reactants or changing external conditions, as usual") or "some very unusual, yet generally applicable larger structures are identified" (is remains unclear what the authors wish to use the compounds for; mind that one won't use "structures", but rather "compounds" anyway). Also the conclusion section ought to be revised in this sense.

There are some differences of opinion regarding the language used and so we have modified certain statements to try to address the reviewer's objection regarding tone. We

hope that the reviewer can appreciate the substantial and careful science presented here; it was not our intention to emphasise any 'buzz' words.

c) The authors talk about "equilibria" or "interconversion" of the species in solution, which I would rather call co-existence in most cases. A real interconversion, i.e., a reaction backwards and forwards, was only (though nicely!) proven for the equilibrium between 2 and 3, whereas in all other cases, different reactant stoichiometries led to different relative amounts of products, which can also happen if their formation pathways are completely independent from each other. I cannot see a reaction herein (besides the mentioned pair of compounds 2 and 3, and the independent study on the formation of 4), where a compound was clearly formed out of another one. If so, this should be said more clearly. If not, some of the equilibrium arrows have to be removed from Figures 1 and 5, and the discussion and conclusion paragraphs need to be adjusted accordingly.

There are 3 distinct equilibrium processes demonstrated in this study, which are correctly illustrated in Figures 1 and 5. The equilibria are:

- $2 \rightleftharpoons 3$
- $4 \rightleftharpoons 2 \rightleftharpoons 3$
- $2 + \text{ZnEt}_2 \rightleftharpoons 1 + 5$

As acknowledged by reviewer 2, we have shown good evidence for the first two processes. In order to clarify the equilibrium between $2 + \text{ZnEt}_2 \rightleftharpoons 1 + 5$, we have performed additional variable temperature NMR experiments. The results clearly show an equilibrium, with reactions proceeding forwards and backwards depending on the conditions (Figs. S41, S42). In the experiment, four equivalents of **2A** were added to 15 equivalents of ZnEt_2 , producing a mixture of **2A**, ZnEt_2 , **5A** and **1A**, as expected. The mixture was monitored by NMR spectroscopy at 298, 308 and 318 K. As the temperature increased the backwards reaction is promoted (19 molecules are converted into 6 in this equilibrium so the backwards reaction is entropically favoured). As a result of this equilibrium, the concentration of complex **2A** increases in the high-temperature ^{31}P NMR spectra.

d) Partial incompleteness of the results regarding the existence and/or mode of identification of the diverse species with different ligands, as well as (in part) different synthetic accesses to some of the compounds, complicates the reading and comprehension of the full study. One might concentrate the report on one series of compounds, or try and complete the other series as well.

Complexes **1-5** are fully characterized in solution and solid state using the ligand DPPA. In support of this study, the closely related ligand $\text{D}^{\text{MeO}}\text{PPA}$ was also used to confirm that the same structures (1, 2, 3 & 5) were obtained and to compare the equilibrium processes $2 \rightleftharpoons 3$ and $2 + \text{ZnEt}_2 \rightleftharpoons 1 + 5$. Using two related ligands, we confirm that there was no 'chance' formation of the complexes and that the cluster nuclearity is generally observed, in one case (**2**) the $\text{D}^{\text{MeO}}\text{PPA}$ version allowed for identification of the solid state structure by X-ray crystallography when it was not possible with DPPA. The same cluster stoichiometries and nuclearities are also observed using DOPA with isolated complexes for 1, 2, 5 (and 3 as a minor species). The flexibility of DOPA is essential for nanoparticle solubility but prevents crystallisation; thus it was necessary to combine the use of DPPA (for structural determination) with the use of DOPA (for nanoparticle synthesis); the equivalence of the clusters in solution, is confirmed by NMR. The only example where complexes with DOPA or $\text{D}^{\text{MeO}}\text{PPA}$ are absent is for compound 4; in accordance with this reviewer's previous comment, we endeavoured to keep discussion of 4 to a minimum.

Minor points of criticism are the following:

a) The crystallographic data of compounds 3A, 4A and 5A suffer from a large number of restraints, which for 5A is dramatic as it exceeds the number of parameters by a factor of >2 here. This should be given another round of X-ray diffraction analyses.

Shift limiting restraints were initially used to aid refinement for 5A; however, once the structure had been sufficiently refined these restraints were no longer necessary. Given the reviewer's concern, these restraints have now been removed (with additional rounds of refinement afterwards) reducing the total restraints from 1995 to 775. All other restraints in structures of 3A, 4A and 5A are applied to more accurately model solvents or disordered phenyl groups. It should also be noted that the data to parameter ratio is greater than 10 in all cases.

b) The rather elaborate report on the nanoparticle synthesis (page 6 of my copy) is dispensable in my view;

We do feel that the nanoparticle synthesis needs to be included so that all readers can understand both the data obtained and the broader implications of the study. We note that other reviewers' requested more information on the nanoparticle synthesis.

it is in most parts a repeat of known literature,

The detailed nanoparticle synthesis and methodology is not a repeat of known literature: the hydrolysis of ZnEt_2 has not been studied before by any in-situ characterization methods. We do state and reference clearly in our manuscript that the 'yellow ZnO phase' has been observed by others during ZnEt_2 hydrolysis. However, it is important to emphasise that there is no characterization or study of these intermediate species reported. It would be difficult for a general reader, not specifically involved in ZnO nanoparticle synthesis by hydrolysis of organometallics, to follow our new data without some experimental framework for context.

the corresponding TEM figures are not at all delighting,

The TEM figures characterizing the 'yellow ZnO material' are not intended to demonstrate the formation of the well-defined nanoparticles which are the final product of the reaction. In fact, the point of including the TEM analysis of the ZnO 'yellow' phase was to demonstrate that the reaction had not yet resulted in the formation of well-defined particles. Perhaps the disordered nature of this phase is why it has never been reported previously. In contrast, once the ligand is coordinated (at the end of the hydrolysis reaction), then very well-dispersed nanoparticles are obtained, as characterised by a further TEM image. Images of the final product are already available in the literature, and it was for this reason that we did not include them in the original version of the manuscript. However, at the request of referee three, we now include an image [Fig. S63] of the final ZnO@DOPA NPs showing a cleanly-resolved equiaxed product.

and the correlation with the series of new compounds remains questionable, as outlined above. One might summarize the findings illustrated in Figure 6 in a short paragraph as an addition to the report on the clusters.

As previously discussed, the yellow ZnO phase grows in parallel with the well defined cluster species which act as a ligand reservoir until the completion of the reaction. We do not claim that structures of the clusters are directly correlated to the nanoparticles.

c) The manuscript contains some inappropriate expressions, such as "pure metal cluster complexes" or "bimetallic clusters" where ligand-bridged complexes or clusters are meant,

We accept this point about nomenclature and have corrected the manuscript:

"pure metal cluster complexes" is changed to "metallic cluster complexes"

In the introduction 'bimetallic' is changed to 'heterobimetallic' which is the term used to describe the complexes in reference 14.

and some typos (c.f. "μ" not printed on page 5).

Thank you – we have corrected the typos.

The second paragraph of the introduction seems to be somewhat unorganized regarding the order of mentioned types of (existing/non-existing) compounds.

We accept this point and have re-ordered the paragraph (p1, 2nd paragraph).

In conclusion, I recommend thorough revision of the work according to the issues raised above. Further, I am not convinced that Nature Communications will be the appropriate platform for communicating the findings, which are definitely nice but presumably not in the position to attract the broad readership of Nature Communications upon focusing on the relevant results. I see it more appropriately published in a chemistry or materials science journal.

Reviewer #3 (Remarks to the Author):

The present article is a very thorough analysis of the chemical processes related to ZnO formation on protolysis/hydrolysis of zinc alkyl compounds. The use of diorganophosphinic acids as stabilizing ligands plays a key role in this study, since various ZnO clusters and also nanoparticles can be supported by this ligand, and the ³¹P NMR signal of the PO₂ group seems to be sensitive enough to its chemical environment allowing a versatile in-situ distinction of different species by ³¹P NMR spectroscopy. In general the work has been performed with great care and the scientific argumentation is sound throughout the whole article. However, some important questions remain unanswered and should be addressed more clearly in my opinion:

1. The in-situ hydrolysis experiment of ZnO NP formation monitored by ¹H and ³¹P NMR spectroscopy is indeed very interesting and most central to the article. In my understanding, the formation of 2 at high water levels, which occurs clearly parallel to the NP formation, is well explained by the fact that 2 contains the highest DOPA:Zn ratio of all identified clusters, i.e. it serves as a "ligand reservoir" of excess DOPA, which is liberated on agglomeration of ZnO units.

We thank the reviewer for this insight with which we entirely agree. The manuscript has been modified accordingly, see p7, right hand paragraph, line 9.

However, by increasing the water level above 100%, I would expect formation of 3, which is not mentioned in the text.

This comment is very sensible except that, in the case of DOPA as the ligand, the equilibrium between **2C**/**3C** lies very much at **2C** (only ~1% of **3C** is present after addition of 6 equivalents of water to **2C**, Fig S48). The position of the equilibrium differs compared to the speciation using DPPA, where the formation of **3A** occurs in much greater percentage. During the specific reactions for the DOPA system, around or above 100% hydrolysis, it is reasonable that **3C** is not observed. We have clarified this in the text, see p7, right hand paragraph, line 12.

Also the formation of free DOPA-H cannot be excluded, which might show a fast ligand exchange with coordinated DOPA preventing its observation in in-situ NMR experiments. Additional experiments, e.g. VT-NMR of DOPA stabilized ZnO NP in presence of excess DOPA-H or the use of deuterated DOPA-D/D₂O are recommended.

The reviewer makes an important point and we agree that the presence or absence of 'free' DOPA-H is relevant. We have undertaken additional DOSY and VT NMR experiments to investigate the solutions containing ZnO@DOPA NPs and **2C** (Figs. S65-S67). Neither experiment show any indication of fast exchange between the ligands coordinated to **2C** and the nanoparticles or presence of free ligand. The DOSY experiment clearly shows two diffusion regimes which are in accordance with the expected hydrodynamic radius of **2C** and the nanoparticles.

We have not undertaken deuterated experiments as reaction of acidic deuterons with ZnEt₂ will form only deuterated ethane. It is also important to (re)emphasise that, in this work, excess DOPA-H is not applied, thereby limiting the utility of deuteration experiments.

2. The reaction scheme shown in Figure 6a is in my opinion misleading, since the stoichiometry presented in the equations is not fully supported by the experiments. I recommend to include the area graph S48 (%P vs %hydrolysis) instead, which is really helpful in understanding the complex interplay of the various chemical processes involved.

We apologise for the original presentation of Fig 6a: the stoichiometry was not clear and there were some errors, as the reviewer points out. The figure has been fully revised; however, we believe the overall scheme is still important to clarify the narrative and data obtained. Nevertheless, we have also moved Fig. S48 into the main manuscript (now part of Fig. 6).

3. ZnO NPs are most often prepared by a sol/gel approach with different types of stabilizing agents, carboxylic acids or carboxylates being amongst the most important ones. The similarities/differences of carboxylates and phosphinates need to be addressed and discussed with more care. E.g. a comparison of the chemical stability of **2** and its well known carboxylate counterpart, especially their hydrolytic stability, might help to convince the reader of the scientific relevance of this study.

This is a critical point which we have improved in our revised manuscript:

Alongside a more detailed comparison of ligands in the introduction on page 1 we now include the following paragraph on p3/4

Given interest in similar carboxylate ligated Zn clusters the different structures and reactivity observed here with phosphinate ligands is notable. The solid state structure of **2B** shows Zn–phosphinate bond lengths ranging from 1.917(2)-1.960(2) Å with an average (1.936(2) Å) slightly greater than that of the analogous Zn-benzoate structure Zn₄O(O₂CPh)₆ (average = 1.926 Å),³⁶ which is in line with slightly weaker bonding from the phosphinate. The Zn–(μ₄-O) bonds in **2B** are also lengthened (average **2B**, 1.989(2)

Å; $\text{Zn}_4\text{O}(\text{O}_2\text{CPh})_6$ 1.946 Å) presumably as a result of the larger size of the phosphinate chelate compared to a carboxylate (average P–O (**2B**), 1.512(2) Å; average C–O ($\text{Zn}_4\text{O}(\text{O}_2\text{CPh})_6$), 1.258 Å),³⁶ which allows for an expansion of the Zn_4O cluster. **2A/B** react with water to form well-defined zinc hydroxide complexes whilst the carboxylate analogue [$\text{Zn}_4\text{O}(\text{CO}_2\text{Ph})_6$] reacts as a Lewis acid towards water to form an aqua complex, [$\text{Zn}_4(\mu_4\text{-O})(\text{OOCPh})_6(\text{H}_2\text{O})(\text{THF})$].³² Thus, complexes with phosphinate ligands undergo disruption of the Zn_4O core and spontaneously rearrange to hexa-zinc tri(hydroxide)clusters. The Zn–OH bonds in **3A** are shorter on average (1.935(2) Å) than the Zn–($\mu_4\text{-O}$) bonds in **2B** (1.989(2) Å), it may be that due to the larger size of the phosphinate ligand effective bonding to the oxo/hydroxo ligand is favoured in the expanded Zn_6 structure.

The sentence "[phosphinates] are more acidic and so stronger binding to zinc is expected" is counterintuitive (phosphinic acids are more acidic than carboxylic acids) and should be discussed/explained.

Once again, we thank the reviewer for noting this point of confusion. Indeed, the pK_a values (diphenylphosphinic acid, 2.3; benzoic acid, 4.2) clearly show that phosphinic acids are more acidic than carboxylic acids. This comparison does mean that the phosphinate would be expected to be slightly weaker ligands, as indicated by the bond distance values observed in the solid state structures. The phosphinate ligands are expected to be more hydrolytically stable compared to carboxylates. We have corrected the manuscript to clarify this point:

See p1, right hand side, lines 23-29

4. While the characterization of the molecular species presented in the article is excellent, the characterization of the ZnO NP is much less conclusive. E.g. TEM images are only given for the "yellow ZnO" but not for the final NP after complete hydrolysis. Why? The results from absorption UV spectroscopy are not convincing, recording of fluorescence spectra is recommended. In general, a more thorough analysis of the final hydrolysis product, i.e. the DOPA capped ZnO NPs, is needed.

We have provided additional contextual detail regarding the nanoparticle synthesis and characterization data (TEM image (Fig. 6c and S63), XRD, IR, UV and photoluminescence spectroscopic data for a typical batch of ZnO@DOPA are illustrated in the ESI Figs. S63-71). We originally included less of this information, as the final ZnO nanoparticle products have been characterised previously, and we wished to focus on the new in situ mechanistic data. However, we can understand that the general reader will wish to see that high quality nanoparticles are indeed formed ultimately, despite the curious ligand behaviour during the synthesis.

5. My major concern is the presentation of the article. While the molecular clusters including the hydrolysis experiment are very thoroughly discussed, the article is less convincing when NP chemistry is concerned. However, the scientific relevance of this study is in my opinion strongly based on its significance for ZnO NP synthesis: Where exactly is the scientific impact of an elucidation of the mechanistic details of ZnO NP formation?

I do not doubt the importance of this study, but a thorough discussion of the state of the art of ZnO NP synthesis, its open questions and problems and discussion of the significance of the present study in the light of these problems and questions would in my opinion strongly enhance the scientific impact of this paper.

Alongside more thorough characterisation of ZnO@DOPA nanoparticles, comparison to ZnO particles prepared without ligand is now included in the discussion. We have extended discussion towards the end of the paper and include the relevant text here (p7,8).

This reaction trajectory is quite unexpected. The highly moisture sensitive alkyl-zinc complex **5C** forms rapidly and is maintained throughout the majority of the hydrolysis reaction, sequestering essentially all the available ligand, whilst residual ZnEt_2 is consumed to form unligated ZnO nanoparticle precursors. A 50% hydrolysed mixture was monitored and found to be unchanged after 15 hours, indicating that the system is in thermodynamic equilibrium, with 93% of the total phosphinate supply incorporated in the form of a stable cluster (minor unidentified species make up the balance to ~100% relative to the internal standard, Figs. 6c, S53). It is only near full hydrolysis of all other Zn–Et species that **5C** reacts with the 'yellow' ZnO precursors to form the phosphinate capped ZnO nanoparticles. Unlike models in the literature, proposing cluster compounds as molecular building blocks that directly map onto the final NP crystal structure,^{63,64} here the cluster compounds do not obviously relate to Wurtzite and instead appear to act only as a reservoir of ligand. This fresh insight into ligand behaviour during nanoparticle synthesis has implications for the concepts of nanoparticle growth and stabilization.

The formation of ZnO nanoparticles by hydrolysis, under the same conditions, but in the absence of any ligand produces insoluble nanoparticles, with average particle sizes of ~3.5 nm (by XRD, Fig. S72). The similarity in size range to the particles prepared using the DOPA ligands (2-3 nm by XRD) indicates that the ligand is not critical for particle size control, in keeping with the distribution of ligands only at the end of the reaction. Previous studies using carboxylate ligands and a similar synthetic protocol also found that ZnO particles were consistently formed within the 3-5 nm size range, regardless of the nature or loading of the ligand applied.¹³ Nonetheless, the ligands are important to produce well-dispersed and soluble nanoparticles as they prevent aggregation observed in their absence. Furthermore, ligands are likely to have a significant impact on subsequent ripening and ageing of the nanoparticles.^{13,20}

This study exploits an organometallic route to nanoparticles that delivers only the stoichiometric quantity of ligand. By avoiding the excess uncoordinated ligand used in many liquid-phase nanoparticle syntheses, the fate of the ligand at various stages of the reaction can be directly determined. Nanoparticle nucleation is very often considered to be a non-equilibrium process, requiring high degrees of super-saturation and high concentrations of active surfactants to minimize the size of the critical nuclei, with particle size controlled by kinetics requiring hot injection, fast mixing, and the like.⁶⁵ Alternatively, sol-gel approaches often involve irreversible condensation reactions.²⁰ The presence of ligands during nanoparticle synthesis is usually assumed to reduce the nucleation barrier and critical nucleus size, by reducing surface energy of the nascent nanoparticle. Smaller particles can therefore form, which are sterically stabilised against coalescence by the coordinated ligand.⁶⁶⁻⁶⁸ Here, this model is completely subverted as the ligands are observed to only interact at the end of the reaction. The behaviour found in this ZnO system may well be observed in other cases, where ligand supply is limited,^{69,70} or in systems with analogous structures, such as (Zn, Cd) (S, Se, Te) capped by coordinating ligands (carboxylates, phosphinates, or phosphonates).

This study shows that equilibrium cluster inter-conversions, including of oxo-bridged species, can play a key role in the distribution of ligand on growing nanoparticles.

Understanding the mechanisms by which the nanoparticle core is formed and then decorated by ligands is likely to help in the formation of (surface) doped nanoparticles crucial for many applications,^{2,71,72} especially in (opto)electronics, and in forming mixed ligand layers which may allow unusual wettability or adaptive behaviour.^{73,74}

In general, I am convinced that the study is highly interesting and the results most relevant to a large chemical community. I recommend acceptance of this article in "Nature Communications" after thorough revision and consideration of the points raised above.

Extra changes to the manuscript

The table of contents graphic has been slightly changed:

CIF for **5A** updated

CIF for **2A** incomplete refinement included for review only

Reviewers' comments:

Reviewer #1 (Remarks to the Author):

The inclusion of PXRD of the bulk material is an important addition to the paper, which, along with the hugely improved reaction yields, addresses the key issues. Lack of solid-state NMR work is a little disappointing, but this shouldn't be an obstacle the paper being accepted.

If the other reviewers are happy that their concerns have been addressed then I recommend that the paper is accepted.

Reviewer #2 (Remarks to the Author):

The manuscript by Shaffer, Williams and co-workers was revised according to most the referees' concerns. As some of the instructions were contradicting, not all of the issues could be addresses, but the authors give reasonable explanations for it.

Overall the clarity of the report was increased such that the crucial points can now be gathered more easily, which was previously not the case.

I can recommend acceptance of the paper for publication upon addressing of a couple of minor points, but I leave the decision to the Editor whether or not the meaningfulness of the reported findings is sufficiently high to merit publication in Nature Communications.

a) The structure of compound 5a should be refined to $ls_shift/su_max = 0.000$ and $ls_shift/su_mean = 0.000$; the Flack parameter should be given.

b) The reply to the first reviewer's request for powder XRD diagrams, i.e. that corresponding study, is not satisfactory so far. The authors essentially prove the identity of compounds 2a and 3a. For 4a, they observe an "amorphous XRD pattern" (contradictive term by the way, as an amorphous material does not produce any XRD pattern), which might be due to too fast precipitation. However, considering the fact that 40% of the material are something else than the product, it again raises the question about the identity of the bulk products. None of the other compounds were commented on regarding powder XRD. This need completion.

c) Whether or not the by-products of the synthesis of 5c are due to imbalances of the reagent stoichiometries and/or traces of oxygen can be easily checked by improving the synthesis conditions. Large-scale synthesis (as done for 1 - 4) helps in the first case, strict exclusion of air and employment d) I don't see a major advantage of the term "building unit" over "seed". The point is that a building unit can be only called that way if its structure is maintained in the final bulk material. Whether or not this is the case in the quoted reports may be checked by the authors themselves. In their case it is definitely not the case, but it was obviously not the authors' intention to come up with it anyway. I like to mention that the introduction is still misleading in this sense, as one expects to see an according story until page 4, when the authors elaborate about the actual "role" of the clusters in their revised paragraph. The sentence written on page 1 "...though it is known that alky zinc alkoxide/carboxylate clusters are useful precursors to ZnO nanoparticles..." is completely in line with this expectation, but should be rather questioned by the authors already here, shouldn't it? If the authors do not see a contradiction here (i.e., for clusters with alkoxide/carboxylate ligands), the statement on the "general applicability" of their own findings and conclusions for ZnO nanoparticle synthesis becomes questionable, in turn. In my view, it would be beneficial for the broad readership to come up with the new picture on the "ligand reservoir" role of the clusters already in the outline - and state to which extent this might apply to other systems apart from the phosphinate-decorated clusters.

e) I still recommend postponing the presentation and discussion of compound 4 to an upcoming paper. The material is complex enough without - and, moreover, it also does without.

Reviewer #3 (Remarks to the Author):

Dear Editor,

the authors have adequately adressed all points raised in my original review . The nanoparticle part has been strengthened and linked to the cluster chemistry by a clever discussion. I recommend acceptance of the article in the current form.

The manuscript by Shaffer, Williams and co-workers was revised according to most the referees' concerns. As some of the instructions were contradicting, not all of the issues could be addresses, but the authors give reasonable explanations for it.

Overall the clarity of the report was increased such that the crucial points can now be gathered more easily, which was previously not the case.

I can recommend acceptance of the paper for publication upon addressing of a couple of minor points, but I leave the decision to the Editor whether or not the meaningfulness of the reported findings is sufficiently high to merit publication in Nature Communications.

a) The structure of compound 5a should be refined to $l_s_shift/su_max = 0.000$ and $l_s_shift/su_mean = 0.000$; the Flack parameter should be given.

b) The reply to the first reviewer's request for powder XRD diagrams, i.e. that corresponding study, is not satisfactory so far. The authors essentially prove the identity of compounds 2a and 3a. For 4a, they observe an "amorphous XRD pattern" (contradictive term by the way, as an amorphous material does not produce any XRD pattern), which might be due to too fast precipitation. However, considering the fact that 40% of the material are something else than the product, it again raises the question about the identity of the bulk products. None of the other compounds were commented on regarding powder XRD. This need completion.

c) Whether or not the by-products of the synthesis of 5c are due to imbalances of the reagent stoichiometries and/or traces of oxygen can be easily checked by improving the synthesis conditions. Large-scale synthesis (as done for 1 - 4) helps in the first case, strict exclusion of air and employment

d) I don't see a major advantage of the term "building unit" over "seed". The point is that a building unit can be only called that way if its structure is maintained in the final bulk material. Whether or not this is the case in the quoted reports may be checked by the authors themselves. In their case it is definitely not the case, but it was obviously not the authors' intention to come up with it anyway. I like to mention that the introduction is still misleading in this sense, as one expects to see an according story until page 4, when the authors elaborate about the actual "role" of the clusters in their revised paragraph. The sentence written on page 1 "...though it is known that alky zinc alkoxide/carboxylate clusters are useful precursors to ZnO nanoparticles..." is completely in line with this expectation, but should be rather questioned by the authors already here, shouldn't it? If the authors do not see a contradiction here (i.e., for clusters with alkoxide/carboxylate ligands), the statement on the "general applicability" of their own findings and conclusions for ZnO nanoparticle synthesis becomes questionable, in turn. In my view, it would be beneficial for the broad readership to come up with the new picture on the "ligand reservoir" role of the clusters already in the outline - and state to which extent this might apply to other systems apart from the phosphinate-decorated clusters.

e) I still recommend postponing the presentation and discussion of compound 4 to an upcoming paper. The material is complex enough without - and, moreover, it also does without.

Response

We thank Reviewer two again for their detailed comments and overall supportive conclusion. We address the additional minor issues below:

a) The structure of **5A** was already reasonably refined at the last submission, with a value for l_s_shift/su_mean of 0.000. To explore this issue, in more detail, the structure of **5A** was further refined. After an additional, 100 cycles l_s_shift/su_max converges to 0.001. However, we emphasize that IUCr checkCIF does not create an alert for values in such a low range (in fact the checkCIF process only creates a c-level alert if this value is above 0.05 ie. ~ 50 times higher than our original data-set).

The Flack parameter was refined (from 5576 Friedel-pairs) and is now given (0.397(10)), it shows that the crystal is composed of two enantiomeric forms of **5A**, as may be expected for quasispherical molecules like **5A**. (*Chirality in transition metal chemistry*, Wiley, H Amouri, M Gruselle). The additionally refined structural data are resubmitted with these corrections.

b) We have now managed to obtain a powder pattern for a bulk sample of the air-sensitive Compound **1A**, which shows excellent agreement with the simulated pattern from the single crystal data. This figure is added to the supporting information.

Fig 1. Powder XRD pattern of **1a**

Preparing bulk samples of the very sensitive compound **5A** continues to prove more problematic. The sensitivity to air and moisture, intrinsic equilibrium character (with volatile $ZnEt_2$ component) limit large scale powder experiments. We originally stated a moderate yield (36%) of pure crystalline material after recrystallisation. This product was analysed by NMR spectroscopy, X-ray crystallography and elemental analysis and in each case found to be of excellent purity (the C

elemental analysis results are within $\pm 0.06\%$ from the calculated value and easily distinguishable from other compounds **1A-4A** by the lower C% in **5A**). We wish to emphasise that good evidence of purity and composition of the new organometallic compound **5A** is reported and that it is not routine to apply powder XRD to characterize such complexes. Reviewer two was not originally concerned about the powder XRD; the initial request from Reviewer one was to distinguish the possibility of dimers/tetramers within the solid-state. However, since **5A** has such a distinctive set of ^1H NMR signals which correspond exactly to the single crystal structure presented (two ethyl groups in a 4:1 ratio), there is sufficient evidence to rule out significant aggregation in this case.

We successfully confirmed the bulk purity of **2A** and **3A** in the first set of corrections, the corresponding complexes **2B** and **3B** (which are very closely analogous to **2A/3A** but feature an ortho-methoxy substituent) are shown to behave identically to the analogous **A** compounds in all forms of spectroscopy or characterization and so further analysis by powder XRD is not necessary to confirm purity or aggregation state. Compounds **1C**, **2C**, **3C** and **5C** are oils and cannot be studied by solid-state methods.

We also note that powder XRD is not a typical characterisation technique for organometallic compounds and is not mentioned in the Nature Communications outline (see footnote) for sample purity confirmation (instead NMR spectroscopy and elemental analysis are requested, as were originally provided)

The 60% isolated yield reported for **4A** does not necessarily mean the remaining 40% is a different compound; as is usual in synthetic chemistry, the yield represents the material collected after the synthetic procedure and, in this case, is mainly limited by material isolation and collection. It is indeed likely that the amorphous nature of **4A** was due to fast precipitation – this fast precipitation method was conducted specifically to remove the recrystallization step, upon request from a different reviewer. Upon slower recrystallization, we obtained the single crystal X-ray structure, mass spectrometry, NMR spectroscopy and elemental analysis data originally reported, note that the NMR spectroscopy data is collected from a much larger volume than a single crystal. X-ray scattering from amorphous materials is very widely studied and has a characteristic appearance.

c) As requested, we have repeated the synthesis of highly sensitive **5C**. The NMR spectrum of the reaction solution directly after synthesis (without any isolation (evacuation) steps) was analysed by ^1H and ^{31}P NMR spectroscopy. This clearly shows the clean formation of **5C** with only a trace impurity of known complex **1C**. The fully assigned spectra are included here and added to the supporting information.

Fig 2. $^{31}\text{P}\{^1\text{H}\}$ NMR spectrum of reaction solution for **5a**. 96% of the ligand is found in the form of **5a**.

Fig 2. ^1H NMR spectrum of reaction solution for **5a**.

d)

The “building block” term was not ours, and was referenced to the work of R. Fischer *et al.* relating to $[\text{Zn}_3\text{Cp}^*_3]^+$ and $\text{Zn}_2\text{CuCp}^*_3$ triangular clusters which are indeed building blocks for metals/alloys. We agree with the reviewer that a distinction should be made between such systems, in which the atomic arrangement in the clusters match the bonding patterns of bulk phases (in that case metallic Zn or brass), and those in which the clusters do not relate obviously to the bulk phase (in our case

Wurzite ZnO). As the reviewer comments, we do not intend to take the reader in this direction, so in order to remove any ambiguity we have removed the phrase including the building block comment.

Deleted – “*and they can serve as ‘building units’ or ‘models’ for bulk materials*”

We clarify that the term precursor is used extensively in the literature (e.g. single source precursor) and does not correspond to a seed or nucleation embryo but only as a reagent.

In previous reports from Boyle and Polarz, the use of $[RZnOR]_4$ ‘cubane’ species are discussed as precursors for ZnO nanomaterials. Whilst the authors comment within the text that “some differences between the precursor and ZnO-Wurzite are also evident” (Ref 12) and “the squares must properly align and then break a single Zn-O bond” (Ref 10), some of their figures demonstrating the joining of cubane units (to form ZnO) can be easily misinterpreted. Therefore to distance our work from these reports and to clarify the direction of our paper we have added the following into the introduction.

The metal-oxygen framework structures of reported precursors do not generally map directly onto the Wurzite-ZnO structure,^{10,12} therefore, their transformations into nanoparticulate ZnO likely involves significant molecular rearrangement. In the case of $[RZnOR]_4$ complexes, alkoxide is lost before ZnO nucleation (to a Wurzite structure) and the relationship of the ligands to the growing nanoparticle is not clear. Here, we show that ligands may coordinate to cluster species which act as spectators whilst ZnO nucleation occurs, and act as a ‘ligand reservoir’ which is only consumed at the end of the synthesis procedure.

We agree with Reviewer two that a previous report proposing that Zn-alkoxide cubanes act as ‘embryos’ for ZnO, must involve significant rearrangement. With this in mind reference 12 has been removed after the following sentence:

Unlike previous reports in the literature, proposing cluster compounds as molecular building blocks that directly map onto the final NP crystal structure,⁶³

e) We maintain our initial response that compound **4a** is useful as a further example of equilibrium processes in these clusters and also shows the potential for introducing further elements into such clusters which could find future use in surface doping of related nanomaterials. We also reiterate that the text commenting on **4a** is brief.

Footnote

The nature communications guidelines states the following upon sample purity:

5. Sample purity

Evidence of sample purity is requested for each new compound. Methods for purity analysis depend on the compound class. For most organic and organometallic compounds, purity may be demonstrated by high-field ^1H NMR or ^{13}C NMR data, although elemental analysis ($\pm 0.4\%$) is encouraged for small molecules. Quantitative analytical methods including chromatographic (GC, HPLC, etc.) or electrophoretic analyses may be used to demonstrate purity for small molecules and polymeric materials.

REVIEWERS' COMMENTS:

Reviewer #2 (Remarks to the Author):

The latest revision of the work by Williams and co-workers and the corresponding explanations are satisfying. The manuscript is ready now for publication in Nature Communications.